# Improving the Trade-off Between Watermark Strength and Speculative Sampling Efficiency for Language Models

**Weiqing He**[*], **Xiang Li**[*], **Li Shen**[†], **Weijie Su**[†], **Qi Long**[†]
University of Pennsylvania
weiqingh@sas.upenn.edu, lx10077@upenn.edu,
{li.shen@pennmedicine, suw@wharton}.upenn.edu, qlong@upenn.edu

## Abstract

Watermarking is a principled approach for tracing the provenance of large language model (LLM) outputs, but its deployment in practice is hindered by inference inefficiency. Speculative sampling accelerates inference, with efficiency improving as the acceptance rate between draft and target models increases. Yet recent work reveals a fundamental trade-off: higher watermark strength reduces acceptance, preventing their simultaneous achievement. We revisit this trade-off and show it is not absolute. We introduce a quantitative measure of watermark strength that governs statistical detectability and is maximized when tokens are deterministic functions of pseudorandom numbers. Using this measure, we fully characterize the trade-off as a constrained optimization problem and derive explicit Pareto curves for two existing watermarking schemes. Finally, we introduce a principled mechanism that injects pseudorandomness into draft-token acceptance, ensuring maximal watermark strength while maintaining speculative sampling efficiency. Experiments further show that this approach improves detectability without sacrificing efficiency. Our findings uncover a principle that unites speculative sampling and watermarking, paving the way for their efficient and practical deployment.[1]

## 1 Introduction

In the era of generative AI, data provenance has become a pressing concern in academia, journalism, and everyday content creation, where ensuring authenticity is both responsible and critical (Weidinger et al., 2022; Starbird, 2019; Milano et al., 2023; Shumailov et al., 2024). Watermarking offers a principled solution: it embeds verifiable signals into generated text by modifying the token sampling process with a recoverable pseudorandom number and a carefully designed sampling strategy (Aaronson, 2023; Kirchenbauer et al., 2023; Kuditipudi et al., 2024). A good watermarking scheme should embed a strong watermark signal, namely the dependence between sampled tokens and pseudorandom numbers. The degree of this dependence, referred to as watermark strength, directly affects detection efficiency (Li et al., 2025;+). However, deploying watermarking in large-scale LLM inference faces two major bottlenecks: tokens are generated sequentially, each requiring a full forward pass, and the lack of parallelization leaves GPUs underutilized. Together, these bottlenecks make watermarking slow and inefficient in real-world deployment.

Speculative sampling addresses these bottlenecks by accelerating autoregressive generation without compromising quality (Chen et al., 2023; Leviathan et al., 2023; Xu et al., 2024). It employs two models: a lightweight draft model that rapidly proposes multiple candidate tokens, and a larger target model that verifies them in parallel. If most draft tokens are accepted, computation is greatly reduced; if not, the target model must regenerate them, negating the speedup. The acceptance process is stochastic, with its probability determined by how closely the draft model's proposals align with the target model's distribution (Yin et al., 2024). Thus, high efficiency requires the two distributions to

---

[*]Equal contribution.
[†]Corresponding authors.

[1]Code is available at https://github.com/hwq0726/watermark-tradeoff.

be sufficiently similar, ensuring a high acceptance rate. Unfortunately, Hu & Huang (2024) show that, when watermarking is combined with speculative sampling, it is impossible to simultaneously achieve both the highest acceptance rate and the strongest watermarking strength—a rather discouraging result that suggests a fundamental trade-off between watermark strength and sampling efficiency.

In this work, we ask whether and how this seemingly unavoidable trade-off can be overcome, trying to pave the way for more efficient deployment of watermarking under speculative sampling. We revisit the impossibility result and identify a potential path forward. A key limitation in (Hu & Huang, 2024) is that watermark strength is defined in a binary manner: watermarking is considered preserved if and only if each token's distribution exactly matches a designated watermarked distribution. This definition, however, doesn't quantify how each token is coupled with a recoverable pseudorandom number. As a result, it overlooks intermediate levels of watermark strength, preventing a nuanced characterization of the trade-off and leaving open the possibility for improvement.

**Contributions.** Building on this observation, we make the following contributions:

- **Quantifying watermark strength.** We introduce a quantitative measure of watermark strength for unbiased watermarks, defined as the expected KL divergence between the watermarked and original token distributions. We show that this measure governs the decay rate of $p$-values, is upper bounded by the entropy of the original distribution, and attains its maximum precisely when tokens are deterministic functions of pseudorandom numbers. Notably, both OpenAI's (Aaronson, 2023) and Google's (Dathathri et al., 2024) watermarking schemes achieve this maximal strength.

- **Characterizing the trade-off.** Based on this measure, we formalize the trade-off curve as the Pareto frontier between watermark strength and (speculative) sampling efficiency. Here, sampling efficiency is quantified by the acceptance rate, following prior work (Hu et al., 2024). For illustration, we show that when both the draft and target models are "linearly" watermarked, this frontier can be characterized by solving a constrained convex optimization problem that maximizes watermark strength subject to a sampling efficiency requirement. Importantly, this formulation is general and can be applied in a plug-and-play manner to any watermarking schemes. As examples, we illustrate the trade-off curves for OpenAI's and Google's watermarking methods.

- **Improving the trade-off.** Finally, we propose a principled mechanism to overcome this trade-off by applying pseudorandom draft-token acceptance. We prove that it achieves maximal watermark strength while preserving speculative sampling efficiency, and empirically verify that it improves detectability under the same efficiency, offering a constructive path toward practical deployment.

**Paper organization.** The remainder of this paper is organized as follows. Section 2 reviews preliminaries on watermarking, speculative sampling, and the previous trade-off. Section 3 introduces a quantitative measure of watermark strength and uses it to fully characterize the trade-off. Section 4 presents our mechanism for improving the trade-off. Section 5 presents experimental results that validate our mechanism. Due to space constraints, we defer the discussion of related work to Appendix A, and provide all proofs in Appendix B and D.

## 2 PRELIMINARIES

For a token $w$ in the vocabulary $\mathcal{W}$, let $\boldsymbol{P}$ denote its distribution. A watermarking scheme can be viewed as a tractable way to modify $\boldsymbol{P}$ using pseudorandomness (Hu et al., 2024). Specifically, it samples $w \sim \boldsymbol{P}_\zeta$ from a modified distribution $\boldsymbol{P}_\zeta := \mathcal{S}(\boldsymbol{P}, \zeta)$, where $\zeta$ is a pseudorandom variable and $\mathcal{S}$ is a carefully designed decoding function. A scheme (or decoder) is said to be unbiased if averaging over pseudorandomness recovers the original distribution, i.e., $\mathbb{E}_\zeta[\boldsymbol{P}_\zeta] = \boldsymbol{P}$. During detection, the task is to decide whether an observed token sequence comes from the original distribution or its watermarked modification. This naturally leads to the hypothesis testing problem:

$$H_0 : w \sim \boldsymbol{P} \text{ and } w \perp \zeta \text{ versus } H_1 : w \sim \boldsymbol{P}_\zeta = \mathcal{S}(\boldsymbol{P}, \zeta). \tag{1}$$

The key idea is to test for statistical dependence between $w$ and the pseudorandom number $\zeta$. Under $H_0$, no watermark is embedded and $w$ is independent of $\zeta$, while under $H_1$ the watermarking mechanism induces a structured dependence. In what follows, we illustrate this framework with two popular watermarking schemes.

**Gumbel-max watermark.** The most influential unbiased watermark is the Gumbel-max watermark (Aaronson, 2023). It is built on the Gumbel-max trick, a widely used sampling method for multinomial

distributions (Gumbel, 1948; Maddison et al., 2014; Jang et al., 2016). The trick generates a set of independent uniform random variables $\zeta = (U_w)_{w \in \mathcal{W}}$ for each token in the vocabulary $\mathcal{W}$, and ensures that $\arg\max_{w \in \mathcal{W}} \frac{\log U_w}{P_w}$ follows the original distribution $\boldsymbol{P} \equiv (P_w)_{w \in \mathcal{W}}$. Building on this observation, Aaronson (2023) proposed the following decoder: $\boldsymbol{P}_\zeta = \mathcal{S}^{\mathrm{gum}}(\boldsymbol{P}, \zeta)$ where

$$(\mathcal{S}^{\mathrm{gum}}(\boldsymbol{P}, \zeta))(w) = \begin{cases} 1, & \text{if } w = \arg\max_{w' \in \mathcal{W}} \frac{\log U_{w'}}{P_{w'}}, \\ 0, & \text{otherwise.} \end{cases} \tag{2}$$

By construction, this watermarking scheme is unbiased (Li et al., 2025).

**SynthID watermark.** The SynthID watermark, proposed by Google (Dathathri et al., 2024), is based on a novel categorical sampling rule called tournament sampling. For a given number of tournament rounds $m$, the pseudorandom numbers is a collection of $m$ random vectors, given by $\zeta = (\boldsymbol{g}_i)_{i=1}^m$, where each $\boldsymbol{g}_i = (g_{i,w})_{w \in \mathcal{W}}$ is a binary vector with entries independently drawn from Bernoulli$(0.5)$. Under the *two-candidate version* of SynthID (the version we use throughout all discussions), the modified distribution can be defined as $\boldsymbol{P}_\zeta = \mathcal{S}^{\mathrm{syn}}(\boldsymbol{P}, \zeta)$, with

$$\mathcal{S}^{\mathrm{syn}}(\boldsymbol{P}, \zeta) = \mathcal{T}_{\boldsymbol{g}_m} \circ \cdots \circ \mathcal{T}_{\boldsymbol{g}_1}(\boldsymbol{P}), \tag{3}$$

where $\mathcal{T}_{\boldsymbol{g}}$ is the operator

$$(\mathcal{T}_{\boldsymbol{g}}(\boldsymbol{P}))(w) = P_w \cdot \left(1 + g_w - \sum_{w': g_{w'}=1} P_{w'}\right). \tag{4}$$

Dathathri et al. (2024) show that $\mathcal{T}_{\boldsymbol{g}}(\boldsymbol{P})$ corresponds to the distribution of the winner in a one-versus-one match: two tokens $w_1, w_2$ are drawn independently from $\boldsymbol{P}$, and the winner is the one with the larger pseudorandom bit value $g_w$. If $g_{w_1} = g_{w_2}$, the tie is broken uniformly at random. Repeating this tournament for $m$ rounds with independent vectors $\boldsymbol{g}_i$ yields $\mathcal{S}^{\mathrm{syn}}(\boldsymbol{P}, \zeta)$ as the distribution of the final winner token.

**Speculative sampling.** Speculative sampling accelerates LLM inference by first drawing a draft token $w'$ from $\boldsymbol{Q}$ and then checking it against the target distribution $\boldsymbol{P}$ (Chen et al., 2023). The draft token is accepted with probability $\min\{1, P_{w'}/Q_{w'}\}$; if it is rejected, a replacement token is sampled from a residual distribution proportional to the excess mass of $\boldsymbol{P}$ over $\boldsymbol{Q}$. This accept/reject process induces a transition kernel on top of $\boldsymbol{Q}$:

$$\mathcal{A}(w|w') = \begin{cases} \min\left(1, \frac{P_w}{Q_w}\right), & \text{if } w' = w, \\ \frac{(P_w - Q_w)_+}{\sum_z (P_z - Q_z)_+} \cdot \left(1 - \frac{P_{w'}}{Q_{w'}}\right)_+, & \text{if } w' \neq w, \end{cases} \tag{5}$$

where $(x)_+ := \max\{x, 0\}$. We denote this kernel as $\mathcal{A}_{\mathrm{spec}}(\boldsymbol{Q}, \boldsymbol{P})$. By construction, applying it on top of $\boldsymbol{Q}$ recovers the target distribution: $\boldsymbol{P} = \mathcal{A}_{\mathrm{spec}}(\boldsymbol{Q}, \boldsymbol{P}) \circ \boldsymbol{Q}$. The acceptance rate under this scheme is $\mathbb{P}(\text{the inital } w' \text{ is not rejected}) = \sum_w \min\{P_w, Q_w\}$, which is the maximum achievable among all kernels that preserve $\boldsymbol{P}$ (see Lemma 3.1).

**Definition 2.1** (Sampling efficiency). *Given a draft $\boldsymbol{Q}_\zeta$ and transition kernel $\mathcal{A}_\zeta$,[2] the sampling efficiency is the expected acceptance rate:*

$$SE(\boldsymbol{Q}_\zeta, \mathcal{A}_\zeta) = \mathbb{E}_\zeta\left[\sum_{w \in \mathcal{W}} \mathcal{A}_\zeta(w|w) Q_{\zeta,w}\right].$$

**An "inevitable" trade-off.** Hu & Huang (2024) prove that speculative sampling cannot simultaneously maintain watermark strength and achieve maximal efficiency. Efficiency is measured by the expected acceptance rate (Def. 2.1), while watermark strength is defined in a binary manner. For any target distribution $\boldsymbol{P}$ and an unbiased decoder $\mathcal{S}$, let $\boldsymbol{P}_\zeta = \mathcal{S}(\boldsymbol{P}, \zeta)$ be the watermarked token distribution. Watermark strength is preserved only if there exists a pair $(\mathcal{S}', \mathcal{A}_\zeta)$ such that $\mathcal{A}_\zeta \circ \boldsymbol{Q}_\zeta = \boldsymbol{P}_\zeta$ exactly for all pairs $(\boldsymbol{Q}, \boldsymbol{P})$ with $\boldsymbol{Q}_\zeta = \mathcal{S}'(\boldsymbol{Q}, \zeta)$. Here, the decoder $\mathcal{S}'$ could be different from $\mathcal{S}$. Under this condition, the sampling efficiency is strictly below the maximum achievable: there must exists a pair $(\boldsymbol{Q}, \boldsymbol{P})$ such that

$$SE(\boldsymbol{Q}_\zeta, \mathcal{A}_\zeta) < \sup_{(\boldsymbol{Q}'_\zeta, \mathcal{A}'_\zeta)} \left\{ SE(\boldsymbol{Q}'_\zeta, \mathcal{A}'_\zeta) : \mathbb{E}_\zeta[\boldsymbol{Q}'_\zeta] = \boldsymbol{Q}, \ \mathbb{E}_\zeta[\mathcal{A}'_\zeta \circ \boldsymbol{Q}'_\zeta] = \boldsymbol{P} \right\}. \tag{6}$$

Conversely, if equality holds in (6) for all pairs $(\boldsymbol{Q}, \boldsymbol{P})$, watermark strength is necessarily lost, i.e., $\mathcal{A}_\zeta \circ \boldsymbol{Q}_\zeta \neq \boldsymbol{P}_\zeta$ for some pair $(\boldsymbol{Q}, \boldsymbol{P})$.

---

[2]We use $\mathcal{A}_\zeta$ to denote a general transition kernel, which is not necessarily dependent on pseudorandomness.

## 3   COMPLETE THE TRADE-OFF CURVE

### 3.1   QUANTIFICATION OF WATERMARK STRENGTH

As introduced in Section 2, prior work (Hu & Huang, 2024) does not fully characterize the trade-off, as it lacks a quantitative notion of watermark strength. Their framework defines strength in a binary way: it is considered preserved only if the actual token distribution exactly matches a designated watermarked distribution. This overlooks the essential idea that strength should capture how strongly each token depends on pseudorandomness, rather than just distributional equivalence. Consequently, intermediate levels of strength cannot be represented, preventing a more nuanced trade-off analysis. Our first contribution is to introduce a quantitative definition of watermark strength, enabling a precise and continuous characterization of this trade-off.

**Definition 3.1.** *For a watermarking scheme that samples tokens from the modified distribution $\boldsymbol{P}_\zeta = \mathcal{S}(\boldsymbol{P}, \zeta)$, its watermark strength is defined as*

$$\mathsf{WS}(\boldsymbol{P}_\zeta) = \mathbb{E}_\zeta[D_{\mathrm{KL}}(\boldsymbol{P}_\zeta \,\|\, \boldsymbol{P})] = \mathbb{E}_\zeta\left[\sum_{w \in \mathcal{W}} P_{\zeta,w} \log \frac{P_{\zeta,w}}{P_w}\right]. \tag{7}$$

*where $D_{\mathrm{KL}}(\boldsymbol{P}_\zeta \| \boldsymbol{P})$ denotes the Kullback-Leibler (KL) divergence between the watermarked distribution $\boldsymbol{P}_\zeta$ and the original distribution $\boldsymbol{P}$. From an information theory perspective, this definition can also be viewed as the conditional KL divergence, and under the unbiasedness condition $\mathbb{E}_\zeta[\boldsymbol{P}_\zeta] = \boldsymbol{P}$, it is equivalent to the mutual information $I(w; \zeta)$.*

**Remark 3.1.** *Watermark strength is conceptually different from the detection efficiency studied in (Li et al., 2025). Watermark strength quantifies the ideal detectability assuming the true token distributions $\boldsymbol{P}_t$ are known, whereas the latter considers worst-case efficiency when each $\boldsymbol{P}_t$ is believed to fall into a prior class without this assumption. As a result, two schemes with comparable watermark strength may still differ in detection efficiency, depending on their sensitivity around the true token distribution. In practice, the Bayesian posterior detection rule in (Dathathri et al., 2024) may help bridge this gap, as it implicitly learns the token distributions from prior data.*

**Interpretation in terms of sample complexity.**   We now explain why the notion of watermark strength in Def. 3.1 is meaningful: it directly quantifies the difficulty of watermark detection. In particular, detection can be formulated as a hypothesis testing problem (cf. Eq. (1)), where the task is to distinguish the original distribution $\boldsymbol{P}$ from its watermarked version $\boldsymbol{P}_\zeta$. Intuitively, greater watermark strength makes this test easier. The next theorem formalizes this intuition by showing that the average watermark strength determines the exponential decay rate of the $p$-value under the uniformly most powerful test (i.e., the likelihood ratio test), and thus the sample complexity required to reach a prescribed significance level.

**Theorem 3.1** (Sample complexity via $p$-value decay)**.** *Let $\alpha \in (0, 1)$ and $w_{1:n} = (w_1, \ldots, w_n)$. Consider the hypothesis testing problem based on $n$ independent samples:*

$$H_0 : w_{1:n} \sim \boldsymbol{P}_1 \otimes \cdots \otimes \boldsymbol{P}_n \text{ with } w_t \perp \zeta_t \ \forall t \quad \textit{versus} \quad H_1 : w_{1:n} \sim \boldsymbol{P}_{1,\zeta_1} \otimes \cdots \otimes \boldsymbol{P}_{n,\zeta_n},$$

*where each $\zeta_t$ is i.i.d., and the log-likelihood ratios $Z_t := \log \frac{\boldsymbol{P}_{t,\zeta_t}(w_t)}{\boldsymbol{P}_t(w_t)}$ are independent, uniformly bounded, and admit a common neighborhood around zero where their moment generating functions are finite. Assume that the average KL divergence converges: $\underline{D} := \lim_{n \to \infty} \frac{1}{n} \sum_{t=1}^n \mathbb{E}_\zeta \left[D_{\mathrm{KL}}(\boldsymbol{P}_{t,\zeta} \| \boldsymbol{P}_t)\right] < \infty$. Then, under $H_1$, the $p$-value of the likelihood ratio test, which is the uniformly most powerful (UMP) test, satisfies*

$$\lim_{n \to \infty} -\frac{1}{n} \log(\text{p-value}) = \underline{D}, \quad \textit{in probability.}$$

*In particular, to guarantee the $p$-value $\leq \alpha$, it is necessary that $n \geq \frac{1}{\underline{D}} \log\left(\frac{1}{\alpha}\right)(1 + o(1))$.*

**Maximum watermark strength.**   A natural question is which watermarking schemes achieve the largest watermark strength, as defined in Def. 3.1. The next theorem shows that for unbiased watermarks, the maximum is attained if and only if the modified token distribution $\boldsymbol{P}_\zeta$ is degenerate—namely, for each pseudorandom number $\zeta$, all probability mass is placed on a single token, so the generated token is a deterministic function of $\zeta$.

**Theorem 3.2** (Maximum watermark strength). *If $\mathbb{E}_\zeta[\boldsymbol{P}_\zeta] = \boldsymbol{P}$, it follows that*

$$\mathsf{WS}(\boldsymbol{P}_\zeta) = \mathrm{Ent}(\boldsymbol{P}) - \mathbb{E}_\zeta[\mathrm{Ent}(\boldsymbol{P}_\zeta)] \le \mathrm{Ent}(\boldsymbol{P}) := -\sum_{w \in \mathcal{W}} P_w \log P_w.$$

*Equality holds if and only if $\mathrm{Ent}(\boldsymbol{P}_\zeta) = 0$ almost surely.*

Interestingly, both the Gumbel-max watermark and the SynthID watermark (in the limit as $m \to \infty$) attain this upper bound.

**Theorem 3.3.** *The Gumbel-max watermark and the SynthID watermark (as $m \to \infty$) achieve the maximum watermark strength in Thm. 3.2.*

### 3.2 TRADE-OFF CURVES AND EXAMPLES

**Formulation of trade-off curves.** Suppose the unwatermarked draft model is $\boldsymbol{Q}$ and the unwatermarked target model is $\boldsymbol{P}$. An unbiased decoder from a family $\mathcal{Q}_{\mathrm{draft}}$ transforms $\boldsymbol{Q}$ into a watermarked draft $\boldsymbol{Q}_\zeta := \mathcal{S}_{\mathrm{draft}}(\boldsymbol{Q}, \zeta)$, where $\zeta$ is a pseudorandom number. A transition kernel $\mathcal{A}_\zeta$ then rectifies $\boldsymbol{Q}_\zeta$ so that the final distribution $\boldsymbol{P}_\zeta := \mathcal{A}_\zeta \circ \boldsymbol{Q}_\zeta$ remains unbiased, i.e., $\mathbb{E}_\zeta[\boldsymbol{P}_\zeta] = \boldsymbol{P}$. With these components in place, we can now introduce the trade-off curve, which is defined in terms of watermark strength (Def. 3.1) and sampling efficiency (Def. 2.1).

**Definition 3.2** (Trade-off curve). *The trade-off curve is a function $T$ that maps an efficiency requirement $r$ (a lower bound on the sampling efficiency) to the largest achievable watermark strength:*

$$L(r) = \max_{\mathcal{S}_{\mathrm{draft}} \in \mathcal{Q}_{\mathrm{draft}},\ \mathcal{A}_\zeta} \mathsf{WS}(\boldsymbol{P}_\zeta) \quad s.t. \quad \boldsymbol{P}_\zeta = \mathcal{A}_\zeta \circ \boldsymbol{Q}_\zeta,\ \mathbb{E}_\zeta[\boldsymbol{P}_\zeta] = \boldsymbol{P},\ \mathsf{SE}(\boldsymbol{Q}_\zeta, \mathcal{A}_\zeta) \ge r.$$

**Lemma 3.1** (Speculative sampling is optimal). *Fix a draft model $\boldsymbol{Q}_\zeta$ and a target model $\boldsymbol{P}_\zeta$, we define the speculative sampling efficiency (SSE) between them as*

$$\mathsf{SSE}(\boldsymbol{Q}_\zeta, \boldsymbol{P}_\zeta) := \sup_{\mathcal{A}_\zeta} \{\mathsf{SE}(\boldsymbol{Q}_\zeta, \mathcal{A}_\zeta) : \boldsymbol{P}_\zeta = \mathcal{A}_\zeta \circ \boldsymbol{Q}_\zeta\} = \mathsf{SE}(\boldsymbol{Q}_\zeta, \mathcal{A}_{\mathrm{spec}}(\boldsymbol{Q}_\zeta, \boldsymbol{P}_\zeta)).$$

*If $\mathbb{E}_\zeta[\boldsymbol{Q}_\zeta] = \boldsymbol{Q}$ and $\mathbb{E}_\zeta[\boldsymbol{P}_\zeta] = \boldsymbol{P}$, then $\mathsf{SSE}(\boldsymbol{Q}_\zeta, \boldsymbol{P}_\zeta) \le 1 - \mathrm{TV}(\boldsymbol{Q}, \boldsymbol{P}) = \mathsf{SSE}(\boldsymbol{Q}, \boldsymbol{P})$.*

As a high level, Def. 3.2 defines the trade-off curve as the Pareto frontier of the achievable region in the plane of watermark strength versus sampling efficiency. Each boundary point gives the strongest watermark attainable under an efficiency requirement $r$. While the definition allows arbitrary kernels $\mathcal{A}_\zeta$, the objective depends only on the induced distribution $\boldsymbol{P}_\zeta = \mathcal{A}_\zeta \circ \boldsymbol{Q}_\zeta$. Lemma 3.1 shows that for any fixed $\boldsymbol{P}_\zeta$, the speculative sampler $\mathcal{A}_{\mathrm{spec}}(\boldsymbol{Q}_\zeta, \boldsymbol{P}_\zeta)$ achieves the maximal efficiency among all $\mathcal{A}_\zeta$ realizing $\boldsymbol{P}_\zeta$. Thus, replacing $\mathcal{A}_\zeta$ with $\mathcal{A}_{\mathrm{spec}}(\boldsymbol{Q}_\zeta, \boldsymbol{P}_\zeta)$ preserves watermark strength and never decreases the efficiency. *Therefore, without loss of generality, the trade-off curve can be studied by restricting to $\mathcal{A}_\zeta = \mathcal{A}_{\mathrm{spec}}(\boldsymbol{Q}_\zeta, \boldsymbol{P}_\zeta)$ and working directly with $\boldsymbol{P}_\zeta$.*

Conceptually, we can view $\boldsymbol{P}_\zeta$ as the output of an unbiased decoder $\mathcal{S}_{\mathrm{target}}$ from a family $\mathcal{Q}_{\mathrm{target}}$ (in parallel to $\boldsymbol{Q}_\zeta$). With this simplification, the trade-off curve can be reformulated as

$$L(r) = \max_{\mathcal{S}_{\mathrm{draft}} \in \mathcal{Q}_{\mathrm{draft}},\ \mathcal{S}_{\mathrm{target}} \in \mathcal{Q}_{\mathrm{target}}} \mathsf{WS}(\boldsymbol{P}_\zeta) \quad s.t. \quad \mathsf{SSE}(\boldsymbol{Q}_\zeta, \boldsymbol{P}_\zeta) \ge r. \tag{8}$$

This formulation is clean and implementation-friendly: once the families $\mathcal{Q}_{\mathrm{draft}}$ and $\mathcal{Q}_{\mathrm{target}}$ are specified, solving (8) yields a concrete visualization of the complete trade-off curve.

**Remark 3.2.** *In many cases, the final distribution $\boldsymbol{P}_\zeta := \mathcal{A}_\zeta \circ \boldsymbol{Q}_\zeta$ admits an explicit closed form. For instance, in (Hu & Huang, 2024), it uses $\boldsymbol{P}_\zeta := \mathcal{A}_{\mathrm{spec}}(\boldsymbol{Q}, \boldsymbol{P}) \circ \boldsymbol{Q}_\zeta$ to achieve the highest sampling efficiency. Such an explicit formula can naturally be regarded as defining a decoder $\mathcal{S}_{\mathrm{target}}$, and all schemes of this type can be collected into the family $\mathcal{Q}_{\mathrm{target}}$.*

**An example trade-off curve.** Now we turn to visualizing the trade-off curve in (8) for two popular watermarking schemes. As discussed earlier, it suffices to specify two families of unbiased decoders for $\boldsymbol{Q}$ and $\boldsymbol{P}$, respectively. To make this concrete, we consider the linearly watermarked classes

$$\mathcal{Q}_{\mathrm{draft}} = \{(1-\theta)\mathrm{Id} + \theta\,\mathcal{S}_{\mathrm{draft}} : \theta \in [0,1]\}, \quad \mathcal{Q}_{\mathrm{target}} = \{(1-\gamma)\mathrm{Id} + \gamma\,\mathcal{S}_{\mathrm{target}} : \gamma \in [0,1]\}, \tag{9}$$

where $\mathrm{Id}$ denotes the identity decoder that leaves the distribution unchanged, and $\mathcal{S}_{\mathrm{draft}}, \mathcal{S}_{\mathrm{target}}$ are prescribed unbiased decoders. In this construction, if we write $\boldsymbol{Q}_\zeta := \mathcal{S}_{\mathrm{draft}}(\boldsymbol{Q}, \zeta)$ and $\boldsymbol{P}_\zeta :=$

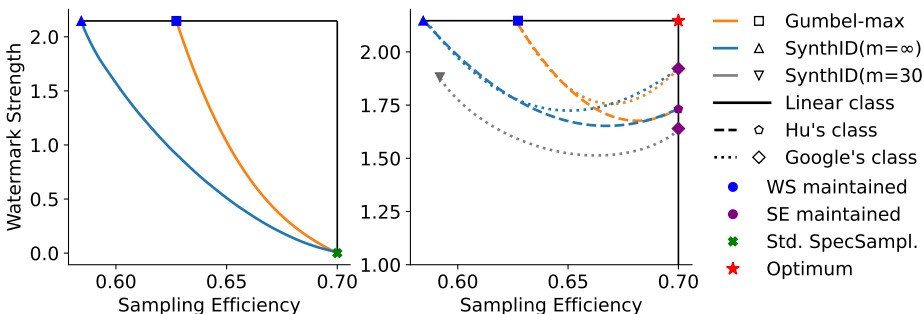

Figure 1: Trade-off curves between watermark strength and sampling efficiency for simulated $(\boldsymbol{Q}, \boldsymbol{P})$ pairs. **Left**: Curves for the linearly watermarked classes (9). **Right**: Curves for other classes, including Hu's class (Hu & Huang, 2024) and Google's class (Dathathri et al., 2024). Orange and blue denote Gumbel-max and SynthID, respectively. Here, solid, dashed, and dotted lines correspond to different classes. Markers indicate the boundary point of each curve.

$\mathcal{S}_{\text{target}}(\boldsymbol{P}, \zeta)$, then identifying the trade-off curve $(r, L(r))$ amounts to finding its inverse curve $(L^{-1}(\rho), \rho)$, where $L^{-1}(\rho)$ is given by

$$L^{-1}(\rho) = 1 - \tfrac{1}{2} \min_{\gamma, \theta} \mathbb{E}_\zeta \|(1-\theta)\boldsymbol{Q} + \theta\boldsymbol{Q}_\zeta - (1-\gamma)\boldsymbol{P} - \gamma\boldsymbol{P}_\zeta\|_1$$

$$\text{s.t.} \quad \mathbb{E}_\zeta\big[\text{Ent}((1-\gamma)\boldsymbol{P} + \gamma\boldsymbol{P}_\zeta)\big] \le \rho. \tag{10}$$

One can derive this formulation (10) by combining Defs. 2.1 and 3.1 with the transition kernel in (5) and the identity $\sum_w \min\{P_w, Q_w\} = 1 - \text{TV}(\boldsymbol{P}, \boldsymbol{Q}) = 1 - \tfrac{1}{2}\|\boldsymbol{P} - \boldsymbol{Q}\|_1$.

The map $(\gamma, \theta) \mapsto \|(1-\theta)\boldsymbol{Q} + \theta\boldsymbol{Q}_\zeta - (1-\gamma)\boldsymbol{P} - \gamma\boldsymbol{P}_\zeta\|_1$ is convex, since it is the $\ell_1$ norm of an affine function. By contrast, entropy is concave, so the feasible set of (10) is not convex in general. Yet when $\mathcal{S}_{\text{target}}$ is degenerate (so $\boldsymbol{P}_\zeta$ is almost surely a point mass), $\mathbb{E}_\zeta[\text{Ent}((1-\gamma)\boldsymbol{P} + \gamma\boldsymbol{P}_\zeta)]$ decreases monotonically in $\gamma$. In this case, the constraint reduces to $\gamma \ge \gamma_0$, where $\gamma_0$ is the unique threshold satisfying $\mathbb{E}_\zeta[\text{Ent}((1-\gamma_0)\boldsymbol{P} + \gamma_0\boldsymbol{P}_\zeta)] = \rho$, and the problem (10) simplifies to

$$L^{-1}(\rho) = 1 - \tfrac{1}{2} \min_{\theta \in [0,1],\, \gamma \in [\gamma_0, 1]} \mathbb{E}_\zeta \|(1-\theta)\boldsymbol{Q} + \theta\boldsymbol{Q}_\zeta - (1-\gamma)\boldsymbol{P} - \gamma\boldsymbol{P}_\zeta\|_1 \,.$$

**Comparisons of trade-off curves.** In Fig. 1, we plot the trade-off curves for simulated $\boldsymbol{Q}$ and $\boldsymbol{P}$ (see Appendix C.1 for the details). The left panel shows the curve for the linearly watermarked classes (9). The green cross at the lower right marks the sampling efficiency of standard speculative sampling, and the two blue points mark the watermark strengths achieved by Gumbel-max and SynthID, respectively. Unless stated otherwise, we set tournament rounds $m = \infty$ for SynthID. As shown in Thm. 3.3, both watermarks attain the same maximal watermark strength, so the blue points lie on the same horizontal line.

The right panel shows trade-off curves for two additional classes: one from Hu & Huang (2024) ("Hu's class") and one from Dathathri et al. (2024) ("Google's class"). While the original works describe how to attain the endpoints of these curves, our framework connects them via a similar linearly interpolated class (see Appendix C.2 for explicit expressions), enabling direct comparison. The results show that *Google's class achieves higher watermark strength than Hu's at matched sampling efficiency, yet neither reaches the theoretical optimum (red star)*. Moreover, when we set $m = 30$—a practical choice for SynthID—and apply it to Google's class, the watermark strength drops below that of Gumbel-max (see the lower gray curve), consistent with Thm. 3.3. This is expected, as the maximal watermark strength is attained only in the limit $m \to \infty$.

## 4 IMPROVING THE TRADE-OFF CURVE

### 4.1 BREAKING THE TRADE-OFF THROUGH PSEUDORANDOM ACCEPTANCE

**Motivation.** With the complete trade-off curve established in Section 3, we now ask whether it can be broken—that is, whether speculative sampling can be used in watermarking to simultaneously

---

**Algorithm 1** Fast watermarked speculative sampling with pseudorandom acceptance

---

1: **Given:** lookahead $K$, output length $N$, target model $\boldsymbol{P}$, draft model $\boldsymbol{Q}$, initial prompt $w_{1:n}$, watermarked models $\boldsymbol{Q}_{\zeta^D} := \mathcal{S}(\boldsymbol{Q}, \zeta^D)$ and $\boldsymbol{P}_{\zeta^T} := \mathcal{S}(\boldsymbol{P}, \zeta^T)$, residual sampler $(\boldsymbol{P} - \boldsymbol{Q})_{+,\zeta^T} := \mathcal{S}((\boldsymbol{P} - \boldsymbol{Q})_+, \zeta^T)$, and pseudorandom generator $G$.
2: **while** $n < N$ **do** ▷ Draft $K$ tokens under the watermarked draft model
3:     **for** $s = 1$ **to** $K$ **do**
4:         Sample draft token $\tilde{w}_s \sim \boldsymbol{Q}_{\zeta^D_{n+s}}(\cdot \mid w_{1:n}, \tilde{w}_{1:s-1})$.
5:     **end for**
6:     **In parallel:** compute $K + 1$ sets of target logits from draft tokens, i.e., $\boldsymbol{P}(\cdot \mid w_{1:n})$, $\boldsymbol{P}(\cdot \mid w_{1:n}, \tilde{w}_1), \ldots, \boldsymbol{P}(\cdot \mid w_{1:n}, \tilde{w}_{1:K})$.
7:     **for** $s = 1$ **to** $K$ **do** ▷ Sequentially try to accept each draft token
8:         Compute pseudorandom $U(0, 1)$ variable: $u_{n+s} \leftarrow G(\zeta^R_{n+s}) \in (0, 1)$.
9:         **if** $u_{n+s} < \min\left\{1, \frac{\boldsymbol{P}(\tilde{w}_s \mid w_{1:n})}{\boldsymbol{Q}(\tilde{w}_s \mid w_{1:n})}\right\}$ **then**
10:             Accept: set $w_{n+1} \leftarrow \tilde{w}_s$;   $n \leftarrow n + 1$.
11:         **else**
12:             Reject: sample $w_{n+1} \sim (\boldsymbol{P} - \boldsymbol{Q})_{+,\zeta^T_n}(\cdot \mid w_{1:n})$;   $n \leftarrow n + 1$; **break**.
13:         **end if**
14:     **end for**
15:     **if all** $\tilde{w}_1, \ldots, \tilde{w}_K$ were accepted **then** ▷ Bonus step as in speculative decoding
16:         Sample one extra token $w_{n+1} \sim \boldsymbol{P}_{\zeta^T_n}(\cdot \mid w_{1:n})$;   $n \leftarrow n + 1$.
17:     **end if**
18: **end while**

---

attain the largest watermark strength and the highest sampling efficiency (SSE). From Lemma 3.1, the maximal SSE for a draft–target pair $(\boldsymbol{Q}, \boldsymbol{P})$ is $1 - \mathrm{TV}(\boldsymbol{Q}, \boldsymbol{P})$. Existing approaches that achieve this bound rely on the transition kernel $\mathcal{A}_{\mathrm{spec}}(\boldsymbol{Q}, \boldsymbol{P})$ in (5), which accepts a draft token $w'$ with probability $\min\{1, P_{w'}/Q_{w'}\}$ (Hu & Huang, 2024; Dathathri et al., 2024). However, this mechanism leaves residual randomness: even with full knowledge of the pseudorandomness in both the watermarked draft and target models, the final token is not predetermined, since it may or may not be the draft token depending on the acceptance coin flip. This inherent randomness weakens watermark strength, because under Def. 2.1, any distribution attaining maximal watermark strength must be degenerate, placing all its mass on a single token. Motivated by this observation, we propose a new approach that preserves both goals: we make the *acceptance decision itself* pseudorandom, so that the entire generation process becomes a deterministic function of pseudorandom variables.

**Algorithm description.** We formally present our method in Alg. 1. The algorithm is driven by a pseudorandom variable with three components $\zeta = (\zeta^D, \zeta^T, \zeta^R)$. The first two components, $\zeta^D$ and $\zeta^T$, determine the watermarked distributions: $\zeta^D$ controls sampling from the draft model $\boldsymbol{Q}_{\zeta^D}$, while $\zeta^T$ controls sampling from the target model $\boldsymbol{P}_{\zeta^T}$ and from the residual distribution $(\boldsymbol{P} - \boldsymbol{Q})_{+,\zeta^T}$ when draft tokens are rejected. The third component, $\zeta^R$, governs acceptance decisions for draft tokens. In particular, at each step $s$, we compute the acceptance variable $u_t = G(\zeta^R_t)$, where $G$ is a pseudorandom number generator producing values uniformly in $[0, 1]$. Additionally, to further ensure the unbiasedness of the entire generated sequence, we apply repeated context masking (Hu et al., 2024; Dathathri et al., 2024; Hu & Huang, 2024) in Alg. 1, which skips watermarking for repeated contexts.

The key difference from (Dathathri et al., 2024) is that the acceptance variable $u$ is now pseudorandom rather than truly random (line 8). As a result, Alg. 1 becomes a fully deterministic function of pseudorandom variables, with no external randomness involved. We show that this modification preserves unbiasedness and, in theory, attains the maximal possible SSE (Thm. 4.1).

**Theorem 4.1.** *Focus on a single intermediate step $s$ and omit the index for brevity. Let $\boldsymbol{P}$ be a target model and $\boldsymbol{Q}$ a draft model. Assume the decoder $\mathcal{S}$ is unbiased and achieves the largest watermark strength (hence it is degenerate by Thm. 3.2). Define the target and draft watermarked distributions by $\boldsymbol{P}_{\zeta^T} = \mathcal{S}(\boldsymbol{P}, \zeta^T)$ and $\boldsymbol{Q}_{\zeta^D} = \mathcal{S}(\boldsymbol{Q}, \zeta^D)$ respectively. Let $\mathcal{A}_\zeta$ denote the transition kernel introduced in Alg. 1, and let $\boldsymbol{P}'_\zeta$ denote the distribution of the output token with $\zeta := (\zeta^D, \zeta^T, \zeta^R)$. Suppose $\zeta^D, \zeta^T$, and $\zeta^R$ are independent. Then the following properties hold:*

(a) **Unbiasedness:** *for every token* $w \in \mathcal{W}$, *we have* $\mathbb{E}_\zeta[\boldsymbol{P}'_\zeta(w)] = \boldsymbol{P}(w)$.

(b) **Maximum sampling efficiency:** $\mathsf{SE}(\boldsymbol{Q}_{\zeta^D}, \mathcal{A}_\zeta) = 1 - \mathrm{TV}(\boldsymbol{Q}, \boldsymbol{P})$.

(c) **Maximum watermark strength:** $\mathsf{WS}(\boldsymbol{P}'_\zeta) = \mathrm{Ent}(\boldsymbol{P})$.

## 4.2 DETECTION UNDER PSEUDORANDOM ACCEPTANCE

Alg. 1 introduces a new pseudorandom component $\zeta^R$, which can also be used to enhance watermark detection. As discussed in Remark 3.1, *watermark strength is conceptually distinct from detection efficiency (detectability)*. While Thm. 4.1 shows that our algorithm preserves the maximum sampling efficiency and attains the maximum watermark strength—thus breaking the trade-off in theory—this does not guarantee optimal detection efficiency. In principle, the most powerful detector is the log-likelihood ratio test, but it is impractical since it requires access to the true token distributions $\boldsymbol{P}_t$ (Huang et al., 2023; Li et al., 2025). Nonetheless, the extra information encoded in $\zeta^R$ reduces uncertainty about the token generation process and can therefore improve detectability. Next, we show how to use $\zeta^R$ to improve detectability for Gumbel-max and SynthID. In Section 5, we provide empirical evidence that our method indeed enhances detection efficiency.

**Gumbel-max watermark.** As described in Section 2, the pseudorandom variables for the Gumbel-max watermark (i.e., $\zeta_t^D, \zeta_t^T$) assign i.i.d. $U(0,1)$ values to all tokens, and the decoder selects the token $w_t$ that solves the argmax in (2). For detection, the corresponding $U(0,1)$ value is extracted as a test statistic (Aaronson, 2023). When watermarked, it tends to concentrate near one; otherwise, it remains uniform. With speculative sampling, however, two candidate statistics arise for each token $w_t$: one from the draft model ($y_t^D \in \mathbb{R}$) and one from the target model or the residual distribution ($y_t^T \in \mathbb{R}$). In our algorithm, $w_t$ comes from the draft model iff $u_t = G(\zeta_t^R) \leq \tau$ (see line 9 in Alg. 1)[3], we naturally select $y_t$ by

$$y_t = y_t^D \mathbf{1}_{G(\zeta_t^R) < \tau} + y_t^T \mathbf{1}_{G(\zeta_t^R) \geq \tau}, \tag{11}$$

where $\tau$ is calibrated on a held-out validation set by grid-searching over candidate values and selecting the one that achieves the highest true positive rate (TPR) under the desired false positive rate (FPR). In contrast, without access to $u_t$, one must rely on the empirical acceptance rate (Dathathri et al., 2024), selecting

$$y_t = y_t^D \text{ with probability } p, \qquad y_t = y_t^T \text{ with probability } 1 - p, \tag{12}$$

where $p$ is estimated from observed acceptance rates. After $y_t$ is chosen, detection proceeds by applying the classic test of Aaronson (2023), which flags watermarking when $\sum_t -\log(1 - y_t)$ is stochastically larger than expected. We refer to the detector based on rule (11) as **Ars-$\tau$** and the one based on rule (12) as **Ars-Prior**. Empirically, the two differ in efficiency.

**SynthID watermark.** A similar challenge of selecting the correct test statistic (i.e., the pseudorandom numbers that generated token $w_t$) also arises in the SynthID watermark. Recall that each pseudorandom variable $\zeta_t^D = (\boldsymbol{g}_{t,i}^D)_{i=1}^m$ consists of $m$ random binary vectors, and the test statistic is defined as $y_t^D = (\boldsymbol{g}_{t,1}^D(w_t), \ldots, \boldsymbol{g}_{t,m}^D(w_t)) \in \mathbb{R}^m$, which collects the $w_t$-th components of all vectors; the counterpart $y_t^T \in \mathbb{R}^m$ is defined analogously. The key detection principle is that, for the correct pseudorandom variable (e.g., $y_t^D$), entries are biased toward one, while for the incorrect one they remain uniformly random in $\{0, 1\}$. This bias stems from the tournament sampling process: since the winning token must repeatedly have larger $g$-values across $m$ rounds, the final $w_t$ tends to carry more ones in its associated vector.

In the prior detection of Dathathri et al. (2024), a Bayesian scoring neural network in $\mathbb{R}^m$ is trained, and under speculative sampling, the two scores from $y_t^D$ and $y_t^T$ are combined through a simple weighted average (see Appendix E for the details). This averaging dilutes the signal and reduces detection efficiency. In contrast, our algorithm has access to the acceptance variable $u_t$, which, while not directly revealing the source model of $w_t$, carries signals about whether the token was generated by the draft or target model. Since the exact threshold (i.e., $\min\{1, P_w/Q_w\}$) separating the two cases is unknown, we treat this as a binary classification problem: given $(y_t^D, y_t^T)$ and $u_t$, a

---

[3]Note that the tokens generated during bonus steps are not controlled by the acceptance variable. However, as long as the lookahead $K$ is not very small (e.g., $K = 1$), the sampling process rarely enters a bonus step, so its impact on detection is negligible in practice.

three-layer perceptron (MLP) is trained to select the correct statistic rather than averaging. We refer to this enhanced method as **Bayes-MLP**, and to the prior approach as **Bayes-Prior**.

## 5 EXPERIMENTS

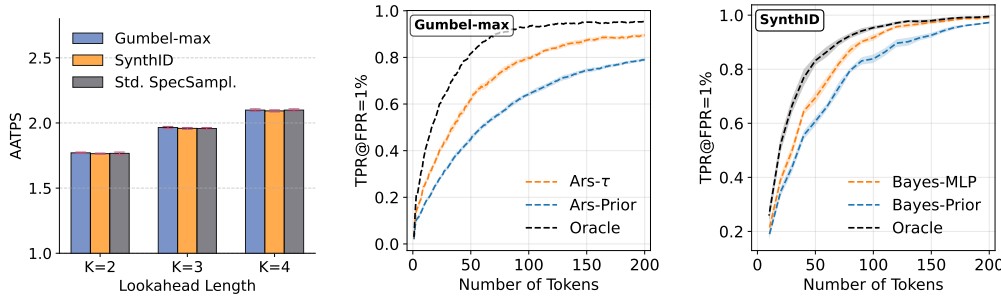

Figure 2: **Left**: Average Accepted Tokens Per Step (AATPS) of Alg. 1 applied to the Gumbel-max and SynthID watermarks, compared with Standard Speculative Sampling (Std. SpecSampl). Error bars mark the 95% confidence intervals. **Middle** and **Right**: Watermark detectability (TPR at FPR = 1%) for Alg. 1 on the Gumbel-max (middle) and SynthID (right). Orange curves show our method, blue curves show the prior-based method, and black curves represent the ideal detector (Oracle) that always selects the correct test statistic. Shaded regions indicate the 95% confidence intervals.

In this section, we show that Alg. 1 simultaneously improves watermark detectability and attains the highest speculative-sampling efficiency. We evaluate Gumbel-max and SynthID ($m = 30$) on the `ELI5` dataset (Fan et al., 2019) for the question-answering task and use two draft–target model pairs: `Llama-68M` & `Llama-7B` (Miao et al., 2024; Touvron et al., 2023) and `Gemma-2B` & `Gemma-7B` (Team et al., 2024). In each pair, the larger model serves as the target and the smaller model as the draft. We report results for the `Llama` pair in the main text and defer the `Gemma` results to the Appendix (see Fig. 3). We compare our methods, **Ars-$\tau$** and **Bayes-MLP**, against the baselines **Ars-Prior** and **Bayes-Prior** (see Section 4.2 for definitions). Besides, we also evaluate Alg. 1 on the `C4` dataset (Raffel et al., 2020) for the open-ended generation task with the same model settings, and the results are provided in Appendix F.2.

Following (Hu & Huang, 2024), we measure sampling efficiency by Average Accepted Tokens per Step (AATPS) from Alg. 1. From the algorithm, at least one token is generated each step, so AATPS lies in $[1, K + 1]$. We report results for $K \in \{2, 3, 4\}$, with larger AATPS indicating higher efficiency. We measure watermark detectability using the true positive rate (TPR) at a fixed false positive rate (FPR) of 1%. To make the results more pronounced, we use lower temperatures: 0.5 for Gumbel-max and 0.7 for SynthID. Since all detection methods in Section 4.2 require training data, each experiment generates 2,000 watermarked texts, split into 1,000 for training and 1,000 for testing. For SynthID, for which the null-score distribution lacks a closed form, we additionally sample two disjoint sets of 1,000 human-written texts from `ELI5` as unwatermarked training and test data.

**Sampling efficiency is maintained.** The left panel of Fig. 2 shows that, for both Gumbel-max and SynthID, Alg. 1 preserves sampling efficiency: the measured AATPS closely matches the standard speculative-sampling baseline. Exact numbers are provided in the Appendix (see Tab. 1 and 2).

**Improved detectability.** The middle and right panels of Fig. 2 show that, for both watermarks, our method attains higher TPR with fewer tokens, demonstrating that pseudorandom acceptance variables enhance watermark detectability. Also, we present the corresponding ROC curves in Fig. 4 and 7, which further corroborate this improvement by providing a more comprehensive characterization of detection performance. Besides, we include an oracle-performance curve representing an ideal detector that always selects the correct test statistic. The results show a gap between our method and this theoretical upper bound—consistent with the analysis in Section 4.2—but the gap is not large, and our method approaches the oracle performance at a token length of 200.

Additionally, we report the Per Token Time (PTT) in milliseconds to evaluate empirical runtime, and the results confirm that Alg. 1 indeed provides acceleration compared to basic unbiased watermarking

methods (without speculative sampling). We also compute the Log Perplexity (LOGPPL) to verify the unbiasedness property of Alg. 1; the results show that it preserves the underlying output distribution and therefore does not degrade the language model's output quality. All results are presented in Tab. 1 and 2.

# 6 CONCLUSION

In this work, we revisited the trade-off between watermark strength and speculative sampling efficiency. We introduced a quantitative notion of watermark strength that links directly to statistical detectability, moving beyond prior binary definitions. With this measure, we cast the trade-off as a constrained optimization problem and derived explicit Pareto curves for existing schemes. Building on these insights, we proposed a principled mechanism that injects pseudorandomness into draft-token acceptance. We proved it achieves maximal watermark strength while preserving speculative sampling efficiency, and we empirically verified improved detectability at the same efficiency.

Our findings suggest several practical directions. First, although we focus on standard speculative sampling, the framework points toward potential extensions to variants such as tree-based methods (Miao et al., 2024; Cai et al., 2024), which could further accelerate generation. Second, while we consider several common decoder classes, future work can explore broader $(\mathcal{Q}_{\text{draft}}, \mathcal{Q}_{\text{target}})$ choices—for example, using different decoders for the draft and target models. Third, our current work directly applies to unbiased degenerate watermarks, but it is an open and interesting direction to investigate how to extend our framework and establish similar improvements to non-degenerate watermarks and even biased ones. In this way, one might have a larger toolbox to trade off generation quality for stronger detection in the context of speculative sampling. Finally, given that human edits can weaken watermark signals (Sadasivan et al., 2023; He et al., 2024a; Li et al., 2025+), investigating the impact of pseudorandom acceptance on robustness to editing is an open direction for future work.

## ACKNOWLEDGMENTS

This work was supported in part by NIH grants R01AG071174, U01CA274576, R01EB036016, R01EB037101, U01AG066833, R01LM014731, and P30AG073105, NSF grant DMS-2310679, a Meta Faculty Research Award, and Wharton AI for Business. The content is solely the responsibility of the authors and does not necessarily represent the official views of the NIH.

## ETHICS STATEMENT

This work studies the interaction between watermarking and speculative sampling in large language models (LLMs). Our research does not involve human subjects or personally identifiable information, and all experiments are conducted using publicly available models and datasets. The purpose of this work is to improve the transparency, traceability, and efficiency of LLM outputs, which we believe aligns with responsible AI development. We are not aware of any foreseeable negative societal impacts from this research.

## REPRODUCIBILITY STATEMENT

We have taken several steps to ensure the reproducibility of our results. All theoretical claims are stated with clear assumptions, and full proofs are provided in Appendix B and D. The experimental setup, including model pairs, datasets, and evaluation protocols, is described in detail in Section 4.2 and 5, with additional details provided in Appendix C, E, and F. To further support reproducibility, we provide anonymized source code for running experiments as supplementary material.

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

## A  RELATED WORK

**Speculative sampling.**  A major challenge in deploying large language models (LLMs) lies in the high inference latency caused by autoregressive decoding, where tokens are generated one by one. Speculative sampling (also referred to as speculative decoding) has emerged as a powerful strategy to mitigate this bottleneck by adopting a draft-then-verify paradigm (Stern et al., 2018; Xia et al., 2023; Leviathan et al., 2023; Chen et al., 2023).

Recent advances have expanded this paradigm through diverse improvements. Some works improve drafting strategies, including independent drafters trained via knowledge distillation (Zhou et al., 2023; Miao et al., 2024; Liu et al., 2024; Kim et al., 2023), and self-drafting methods that reuse parts of the target model, such as Medusa (Cai et al., 2024) and EAGLE (Li et al., 2024). Others explore verification mechanisms, extending beyond single-candidate check to multi-candidates or tree-structured verification (Yang et al., 2024; Miao et al., 2024; Spector & Re, 2023; Sun et al., 2023; He et al., 2024b; Cai et al., 2024; Li et al., 2024). Collectively, these innovations have produced Pareto improvements in both acceptance rate and throughput (Xia et al., 2024).

Our work builds on this growing body of research but explores a new dimension: how speculative sampling interacts with watermarking. Thus, in this work, we focus on the basic speculative sampling method. Importantly, our approach does not rely on any assumptions about the draft or target models, which means the underlying idea can naturally extend to more advanced variants.

**Watermarking techniques.**  Recent studies have proposed a diverse set of watermarking schemes (Kirchenbauer et al., 2024; Fernandez et al., 2023; Kuditipudi et al., 2024; Hu et al., 2024; Wu et al., 2024; Zhao et al., 2025; 2024; Liu & Bu, 2024; Giboulot & Teddy, 2024; Fu et al., 2024; Xie et al., 2025; Dathathri et al., 2024; He et al., 2025a). Most approaches operate by introducing pseudorandomness into the next-token prediction process, so that the randomness—once seeded—creates statistical patterns that can later be detected to verify the presence of a watermark. Besides, recent works have also studied watermarking from an information-theoretic and optimization-based perspective to design stronger watermark schemes (Tsur et al., 2025a;b). Importantly, a watermarking decoder is called unbiased when its token distribution remains identical to the underlying token distribution (He et al., 2025b; Li et al., 2025).

Our work focuses on the unbiased watermark and formally quantifies the watermark strength. Building on this foundation, we revisit the trade-off between sampling efficiency and watermark strength identified by Hu & Huang (2024), and improve this trade-off by extending the one-dimensional pseudorandom seed in traditional watermarks into a multi-dimensional design.

## B  PROOF FOR SECTION 3

### B.1  PROOF THEOREM 3.1

*Proof of Theorem 3.1.*  Under $H_1$, the random variables $Z_1, \ldots, Z_n$ are independent, with $\mathbb{E}_{H_1}[Z_t] = D_t := \mathbb{E}_{\zeta_t}[D_{\mathrm{KL}}(\boldsymbol{P}_{t,\zeta_t} \| \boldsymbol{P}_t)]$. Let $\Lambda_n := \sum_{t=1}^{n} Z_t$ and define the empirical average $\bar{D}_n := \frac{1}{n}\sum_{t=1}^{n} D_t$. By assumption, $\bar{D}_n \to \underline{D}$ as $n \to \infty$. Since the $Z_t$ are independent with bounded moments, the weak law of large numbers gives:

$$\frac{1}{n}\Lambda_n \xrightarrow{P} \underline{D}, \quad \text{under } H_1.$$

Let $\Lambda_n^{\mathrm{obs}} := \Lambda_n$ denote the observed log-likelihood ratio under $H_1$, and define the $p$-value as

$$\mathrm{pval}_n := \mathbb{P}_{H_0}(\Lambda_n \geq \Lambda_n^{\mathrm{obs}}).$$

To evaluate this, we apply the non-i.i.d. version of Cramér's theorem (e.g., Section 2.6 in (Durrett, 2013)). Define the averaged log moment generating function under $H_0$:

$$\psi_n(s) := \frac{1}{n}\sum_{t=1}^{n} \log \mathbb{E}_{H_0}[e^{sZ_t}],$$

and the corresponding rate function:

$$I_n(x) := \sup_{s \in \mathbb{R}} \left(sx - \psi_n(s)\right).$$

**Lemma B.1** (Uniform control of log-MGFs). *Let $\{Z_t\}_{t=1}^n$ be independent random variables such that for some constant $M > 0$, each $Z_t$ satisfies $|Z_t| \le M$ almost surely. Define the log-moment generating functions $\psi_t(s) := \log \mathbb{E}[e^{sZ_t}]$ and their average $\psi_n(s) := \frac{1}{n}\sum_{t=1}^n \psi_t(s)$. Then:*

1. *For any compact interval $K \subset \mathbb{R}$, we have $\sup_{t \le n,\, s \in K} |\psi_t(s)| < \infty$.*

2. *$\psi_n(s)$ converges uniformly on compact intervals to a limiting convex function $\psi(s)$.*

3. *The corresponding sequence of convex conjugates $I_n(x) := \sup_{s \in \mathbb{R}}\{sx - \psi_n(s)\}$ converges pointwise to $I(x) := \sup_{s \in \mathbb{R}}\{sx - \psi(s)\}$.*

By Lemma B.1, the sequence $\{\psi_n(s)\}$ of averaged log-MGFs is uniformly controlled, and the corresponding rate functions $I_n(\cdot)$ converge pointwise to a limiting convex function $I(\cdot)$. Hence, the non-i.i.d. version of Cramér's theorem applies to the sum $\Lambda_n := \sum_{t=1}^n Z_t$, and a large deviation principle holds with rate function $I(\cdot)$.

Since $\Lambda_n^{\text{obs}} = n\underline{D} + o_p(n)$ under $H_1$, the large deviation estimate gives:

$$\mathbb{P}_{H_0}(\Lambda_n \ge \Lambda_n^{\text{obs}}) = \exp(-nI_n(\underline{D}) + o(n)).$$

By Lemma B.1, we have $I_n(\underline{D}) \to I(\underline{D})$ as $n \to \infty$. Moreover, because each $Z_t$ is a log-likelihood ratio satisfying $\mathbb{E}_{H_0}[e^{Z_t}] = 1$, the rate function achieves its maximum at $s = 1$, implying that:

$$I(\underline{D}) = \underline{D}.$$

Therefore,

$$-\frac{1}{n} \log \text{pval}_n \to \underline{D} \quad \text{in probability under } H_1,$$

which implies that

$$\text{pval}_n = \exp(-n\underline{D} + o(n)).$$

To guarantee $\text{pval}_n \le \alpha$, it is necessary that

$$\exp(-n\underline{D} + o(n)) \le \alpha \quad \Rightarrow \quad n \ge \frac{1}{\underline{D}} \log\left(\frac{1}{\alpha}\right)(1 + o(1)).$$

This completes the proof.

$\square$

We conclude by providing the proof of Lemma B.1 below.

*Proof of Lemma B.1.* Since $|Z_t| \le M$ almost surely, for any $s \in \mathbb{R}$ we have

$$e^{-M|s|} \le e^{sZ_t} \le e^{M|s|},$$

which implies that the moment generating function $\mathbb{E}[e^{sZ_t}]$ exists and is bounded by $e^{M|s|}$ for all $t$ and $s \in \mathbb{R}$. In particular, for any compact interval $K \subset \mathbb{R}$, there exists a constant $C_K < \infty$ such that

$$\sup_{t \le n,\, s \in K} |\psi_t(s)| = \sup_{t \le n,\, s \in K} \left|\log \mathbb{E}[e^{sZ_t}]\right| \le C_K.$$

This proves the first part.

Next, observe that each $\psi_t(s)$ is convex (as the log of an MGF) and differentiable. Moreover, since $|Z_t| \le M$, we have

$$\left|\frac{d}{ds}\psi_t(s)\right| = \left|\frac{\mathbb{E}[Z_t e^{sZ_t}]}{\mathbb{E}[e^{sZ_t}]}\right| \le M,$$

so each $\psi_t(s)$ is Lipschitz continuous with Lipschitz constant at most $M$ on all of $\mathbb{R}$. Hence, the sequence $\psi_n(s)$ is equicontinuous and uniformly bounded on compact sets. By the Arzelà–Ascoli theorem, the sequence $\psi_n(s)$ has a uniformly convergent subsequence on each compact interval. Since the pointwise limit

$$\psi(s) := \lim_{n \to \infty} \psi_n(s)$$

exists by the law of large numbers, we conclude that the convergence is in fact uniform on compacts. This proves the second part.

Finally, since each $\psi_n(s)$ is convex and converges uniformly on compact sets to a convex function $\psi(s)$, the corresponding convex conjugates (rate functions) $I_n(x) := \sup_{s \in \mathbb{R}} \{sx - \psi_n(s)\}$ converge pointwise to $I(x) := \sup_{s \in \mathbb{R}} \{sx - \psi(s)\}$ by standard results from convex analysis (e.g., Rockafellar's theorem on epi-convergence of convex conjugates (Rockafellar, 1997)). This proves the last part. □

### B.2 PROOF OF THEOREM 3.2

*Proof of Theorem 3.2.* By definition,

$$\mathsf{WS}(\boldsymbol{P}_\zeta) = \mathbb{E}_\zeta[D_{\mathrm{KL}}(\boldsymbol{P}_\zeta \| \boldsymbol{P})] = \mathbb{E}_\zeta \sum_w P_{w,\zeta} \log \frac{P_{w,\zeta}}{P_w} = \mathrm{Ent}(\boldsymbol{P}) - \mathbb{E}_\zeta[\mathrm{Ent}(\boldsymbol{P}_\zeta)] \leq \mathrm{Ent}(\boldsymbol{P}).$$

When the equality holds, we must have $\mathbb{E}_\zeta[\mathrm{Ent}(\boldsymbol{P}_\zeta)] = 0$ so that $\mathrm{Ent}(\boldsymbol{P}_\zeta) = 0$ for any $\zeta$. □

### B.3 PROOF OF THEOREM 3.3

*Proof of Theorem 3.3.* The decoder for the Gumbel-max watermark is deterministic and always produces a degenerate distribution. Therefore, it trivially achieves the maximum watermark strength.

We now prove the result for the SynthID watermark. Recall that the $m$-layer decoder is defined in (3):

$$\mathcal{S}^{\mathrm{syn}}(\boldsymbol{P}, \zeta) = \mathcal{T}_{\boldsymbol{g}_m} \circ \cdots \circ \mathcal{T}_{\boldsymbol{g}_1}(\boldsymbol{P}), \tag{3}$$

where each $\mathcal{T}_{\boldsymbol{g}}$ is a vectorized operator defined by

$$(\mathcal{T}_{\boldsymbol{g}}(\boldsymbol{P}))(w) = P_w \cdot \left( 1 + g_w - \sum_{w':g_{w'}=1} P_{w'} \right). \tag{4}$$

A direct calculation shows that this transformation preserves expectation:

$$\mathbb{E}_{\boldsymbol{g}}[\mathcal{T}_{\boldsymbol{g}}(\boldsymbol{P})] = \boldsymbol{P}. \tag{13}$$

Let us define

$$\hat{\boldsymbol{P}}_t := \mathcal{T}_{\boldsymbol{g}_t} \circ \cdots \circ \mathcal{T}_{\boldsymbol{g}_1}(\boldsymbol{P}),$$

and let $\mathcal{F}_t := \sigma(\{\boldsymbol{g}_l\}_{l=1}^t)$ be the sigma-field generated by all pseudorandom masks up to layer $t$. By construction and the unbiasedness in (13), we have

$$\mathbb{E}[\hat{\boldsymbol{P}}_t \mid \mathcal{F}_{t-1}] = \hat{\boldsymbol{P}}_{t-1}.$$

This shows that the sequence $\{\hat{\boldsymbol{P}}_t\}$ forms a non-negative, vector-valued martingale adapted to the filtration $\{\mathcal{F}_t\}$. Moreover, each $\hat{\boldsymbol{P}}_t$ is a valid categorical distribution.

By the martingale convergence theorem (e.g., (Durrett, 2013, Section 5.2)), the sequence $\hat{\boldsymbol{P}}_t$ converges almost surely to a limiting distribution, which we denote by $\hat{\boldsymbol{P}}$. We assert that $\hat{\boldsymbol{P}}$ must be a fixed point of the operator $\mathcal{T}_{\boldsymbol{g}}$ for every possible value of $\boldsymbol{g}$ due to the above almost sure convergence. By the definition in (4), this is only possible if $\hat{\boldsymbol{P}}$ assigns all mass to a single token—that is, $\hat{\boldsymbol{P}}$ is degenerate. If this were not the case, then applying $\mathcal{T}_{\boldsymbol{g}}$ would change the distribution for some choices of $\boldsymbol{g}$. Therefore, the SynthID decoder also converges to a degenerate distribution and achieves maximum watermark strength. This completes the proof. □

### B.4 PROOF OF LEMMA 3.1

*Proof of Lemma 3.1.* We first prove the first part. For any fixed $\zeta$, we can write

$$P_{\zeta,w'} = \sum_{w \in \mathcal{W}} \mathcal{A}_\zeta(w' \mid w) Q_{\zeta,w}.$$

In particular, this implies

$$P_{\zeta,w'} \geq \mathcal{A}_\zeta(w' \mid w') Q_{\zeta,w'}$$

$$\Rightarrow \mathcal{A}_\zeta(w' \mid w') \le \frac{P_{\zeta,w'}}{Q_{\zeta,w'}}$$

$$\Rightarrow \mathcal{A}_\zeta(w' \mid w')Q_{\zeta,w'} \le Q_{\zeta,w'} \cdot \min\left(1, \frac{P_{\zeta,w'}}{Q_{\zeta,w'}}\right).$$

Summing over $w' \in \mathcal{W}$, we obtain

$$\sum_{w' \in \mathcal{W}} \mathcal{A}_\zeta(w' \mid w')Q_{\zeta,w'} \le \sum_{w' \in \mathcal{W}} Q_{\zeta,w'} \cdot \min\left(1, \frac{P_{\zeta,w'}}{Q_{\zeta,w'}}\right) = \sum_{w \in \mathcal{W}} \min\left(Q_{\zeta,w}, P_{\zeta,w}\right).$$

Therefore, the sampling efficiency satisfies

$$\mathsf{SE}(\boldsymbol{Q}_\zeta, \mathcal{A}_\zeta) \le \mathbb{E}_\zeta\left[\sum_{w \in \mathcal{W}} \min\left(Q_{\zeta,w}, P_{\zeta,w}\right)\right].$$

Finally, note that equality holds when $\mathcal{A}_\zeta = \mathcal{A}_{\mathrm{spec}}(\boldsymbol{Q}_\zeta, \boldsymbol{P}_\zeta)$, completing the proof of the first part.

For the second part, we note that the total variation distance satisfies the following characterization:

$$\mathrm{TV}(\boldsymbol{Q}, \boldsymbol{P}) = \inf\{\mathbb{P}(X \ne Y) : X \sim \boldsymbol{Q}, Y \sim \boldsymbol{P}\},$$

where the infimum is taken over all couplings $(X, Y)$ such that their marginal distributions are $\boldsymbol{Q}$ and $\boldsymbol{P}$, respectively. Let $\mathcal{A}_\zeta = \mathcal{A}_{\mathrm{spec}}(\boldsymbol{Q}_\zeta, \boldsymbol{P}_\zeta)$. Since $(X, Y) \sim (\boldsymbol{Q}_\zeta, \mathcal{A}_\zeta \circ \boldsymbol{Q}_\zeta)$ forms a valid coupling of $\boldsymbol{Q}$ and $\boldsymbol{P}$,

$$\begin{aligned}
\mathrm{TV}(\boldsymbol{Q}, \boldsymbol{P}) &\le \mathbb{P}_\zeta(X \ne Y; (X, Y) \sim (\boldsymbol{Q}_\zeta, \mathcal{A}_\zeta \circ \boldsymbol{Q}_\zeta)) \\
&= 1 - \mathbb{E}_\zeta\Big[\sum_{w \in \mathcal{W}} \mathcal{A}_\zeta(w|w)Q_{\zeta,w}\Big] \\
&= 1 - \mathsf{SE}(\boldsymbol{Q}_\zeta, \mathcal{A}_{\mathrm{spec}}(\boldsymbol{Q}_\zeta, \boldsymbol{P}_\zeta)) \\
&= 1 - \mathsf{SSE}(\boldsymbol{Q}_\zeta, \boldsymbol{P}_\zeta).
\end{aligned}$$

$\square$

## C  EXAMPLES OF TRADE-OFF CURVES

### C.1  SIMULATION SETUP

To get the numerical result of the trade-off curves,we manually specify 10-dimensional token distributions for the draft and target models:

$$\boldsymbol{Q} = (0.4, 0.10, 0.12, 0.11, 0.08, 0.06, 0.05, 0.035, 0.025, 0.02)$$
$$\boldsymbol{P} = (0.1, 0.13, 0.155, 0.115, 0.235, 0.065, 0.055, 0.05, 0.06, 0.035)$$

These distributions mimic a common pattern observed in practice: the draft model $\boldsymbol{Q}$ concentrates more probability mass on a single token, whereas the target model $\boldsymbol{P}$ exhibits higher entropy. Although the actual vocabulary size in LLMs is far larger than 10, in practice most of the probability mass is concentrated on a small set of high-probability tokens. This is consistent with the intuition behind top-$k$ sampling, where only a handful of tokens dominate the distribution. Thus, while simplified, our simulation setting captures the essential structure of real-world scenarios.

To approximate expectations without a simple closed-form expression (e.g., sampling efficiency), we employ Monte Carlo estimation using $10^7$ pseudorandom seeds and report the resulting empirical mean.

**Implementation of SynthID ($m = \infty$).** According to Thm. 3.3, the SynthID decoder converges almost surely to a degenerate distribution as the tournament rounds $m \to \infty$. In practice, however, we cannot really set $m$ to infinity. Empirically, we observe that at $m = 30$, the distribution is already highly concentrated on a single token, though not yet fully degenerate. By the convergence guarantee, the remaining probability mass will collapse onto this token as $m$ increases further. Thus, in implementation, we approximate the limit distribution by constructing a one-hot vector for that token.

## C.2 OTHER WATERMARKED CLASSES

In Section 3.2, we used the linearly watermarked classes (9) as a simple example to illustrate trade-off curves. More generally, different choices of the watermarked classes $\mathcal{Q}_{\mathrm{draft}}$ and $\mathcal{Q}_{\mathrm{target}}$ yield different curves. These choices can be highly customized, highlighting the flexibility and scalability of our framework.

Here, we detail the classes used in the right part of Fig. 1. We begin with the transition kernel from (Dathathri et al., 2024):

$$\mathcal{A}_\xi(w|w') = \begin{cases} \min\left(1, \frac{P_w}{Q_w}\right), & \text{if } w' = w, \\ \mathcal{S}\big((\boldsymbol{P}-\boldsymbol{Q})_+, \xi\big)(w) \cdot \left(1 - \frac{P_{w'}}{Q_{w'}}\right)_+, & \text{if } w' \neq w, \end{cases} \tag{14}$$

where $(\boldsymbol{P}-\boldsymbol{Q})_+$ denotes the normalized excess mass of $\boldsymbol{P}$ over $\boldsymbol{Q}$ and $\mathcal{S}$ is an unbiased decoder. We denote this kernel as $\mathcal{A}_\xi(\boldsymbol{Q}, \boldsymbol{P})$. Importantly, given the independence of $\zeta$ and $\xi$, we have $\mathbb{E}_{\zeta,\xi}[\mathcal{A}_\xi(\boldsymbol{Q},\boldsymbol{P}) \circ \boldsymbol{Q}_\zeta] = \boldsymbol{P}$ (Dathathri et al., 2024). By Remark 3.2, this transformation can be treated as an unbiased decoder, denoted $\mathcal{S}_{\mathrm{google}}(\boldsymbol{P}, \zeta, \xi) := \mathcal{A}_\xi(\boldsymbol{Q},\boldsymbol{P}) \circ \boldsymbol{Q}_\zeta$. We then define **Google's class** (Dathathri et al., 2024) as:

$$\mathcal{Q}_{\mathrm{draft}} = \{\mathcal{S}_{\mathrm{draft}}\}, \quad \mathcal{Q}_{\mathrm{target}} = \{(1-\gamma)\mathcal{S}_{\mathrm{google}} + \gamma \mathcal{S}_{\mathrm{target}} : \gamma \in [0,1]\}.$$

Similarly, we denote $\mathcal{S}_{\mathrm{hu}}(\boldsymbol{P}, \zeta) := \mathcal{A}_{\mathrm{spec}}(\boldsymbol{Q}, \boldsymbol{P}) \circ \boldsymbol{Q}_\zeta$ and define **Hu's class** (Hu & Huang, 2024) as:

$$\mathcal{Q}_{\mathrm{draft}} = \{\mathcal{S}_{\mathrm{draft}}\}, \quad \mathcal{Q}_{\mathrm{target}} = \{(1-\gamma)\mathcal{S}_{\mathrm{hu}} + \gamma \mathcal{S}_{\mathrm{target}} : \gamma \in [0,1]\}.$$

In both cases, $\mathcal{S}_{\mathrm{draft}}$ and $\mathcal{S}_{\mathrm{target}}$ denote prescribed unbiased decoders (e.g., $\mathcal{S}^{\mathrm{gum}}$ or $\mathcal{S}^{\mathrm{syn}}$ in our experiments). In the above definition, the draft decoder is fixed, and the target decoder is controlled solely by the variable $\gamma$. Hence, by simply iterating over $\gamma$ and applying Monte Carlo estimation, we can compute the sampling efficiency and watermark strength for each value and plot the corresponding trade-off curve.

## D PROOF OF THEOREM 4.1

*Proof of Theorem 4.1.* We first show (a). For Alg. 1, tokens can be generated in two cases. **Case 1**: The token is sampled from the accept and reject loop on lines 7 to 14. By definition,

$$\boldsymbol{P}'_\zeta(w) = \boldsymbol{Q}_{\zeta^D}(w)\,\mathbf{1}_{G(\zeta^R)<\min\{1,\frac{\boldsymbol{P}(w)}{\boldsymbol{Q}(w)}\}}$$

$$+ \left(1 - \sum_{w\in\mathcal{W}} \boldsymbol{Q}_{\zeta^D}(w)\,\mathbf{1}_{G(\zeta^R)<\min\{1,\frac{\boldsymbol{P}(w)}{\boldsymbol{Q}(w)}\}}\right)(\boldsymbol{P}-\boldsymbol{Q})_{+,\zeta^T}(w). \tag{15}$$

The first term corresponds to the probability of sampling $w$ from the draft model and accepting it. The second term corresponds to the probability of rejecting any token from the draft model, then sampling $w$ from the residual distribution. Since the watermark decoder $\mathcal{S}$ is unbiased, we have,

$$\mathbb{E}[\boldsymbol{Q}_{\zeta^D}] = \boldsymbol{Q}, \mathbb{E}[\boldsymbol{P}_{\zeta^T}] = \boldsymbol{P}, \text{and } \mathbb{E}[(\boldsymbol{P}-\boldsymbol{Q})_{+,\zeta^T}] = (\boldsymbol{P}-\boldsymbol{Q})_+.$$

By the definition of $G(\zeta^R)$, we also have,

$$\mathbb{E}[\mathbf{1}_{G(\zeta^R)<\min\{1,\frac{\boldsymbol{P}(w)}{\boldsymbol{Q}(w)}\}}] = \min\left\{1, \frac{\boldsymbol{P}(w)}{\boldsymbol{Q}(w)}\right\}.$$

Since $\zeta^D, \zeta^T$, and $\zeta^R$ are independent, it then follows that,

$$\mathbb{E}_{\zeta=(\zeta^D,\zeta^T,\zeta^R)}[\boldsymbol{P}'_\zeta(w)] = \boldsymbol{Q}(w)\,\min\left\{1, \frac{\boldsymbol{P}(w)}{\boldsymbol{Q}(w)}\right\}$$

$$+ \left(1 - \sum_{w\in\mathcal{W}} \boldsymbol{Q}(w)\,\min\left\{1, \frac{\boldsymbol{P}(w)}{\boldsymbol{Q}(w)}\right\}\right)(\boldsymbol{P}-\boldsymbol{Q})_+(w).$$

This expression is equal to the probability distribution of the next token generated by standard speculative sampling, and Chen et al. (2023) has shown it is equal to the target distribution $\boldsymbol{P}(w)$. **Case 2**: The token is sampled from $\boldsymbol{P}_{\zeta^T}$ (e.g., the bonus step on line 16). Since $\mathcal{S}$ is unbiased, we also have $\mathbb{E}[\boldsymbol{P}_{\zeta^T}] = \boldsymbol{P}$. Thus we prove the unbiasedness.

We then show (b). From Alg. 1, we find that the self-transition probability is $\mathcal{A}_\zeta(w \mid w) = \mathbf{1}_{G(\zeta^R) < \min\{1, \boldsymbol{P}(w)/\boldsymbol{Q}(w)\}}$. Hence, by Def. 2.1,

$$
\begin{aligned}
\mathsf{SE}(\boldsymbol{Q}_{\zeta^D}, \mathcal{A}_\zeta) &= \mathbb{E}_{\zeta = (\zeta^D, \zeta^T, \zeta^R)}\big[\sum_{w \in \mathcal{W}} \mathcal{A}_\zeta(w|w)\boldsymbol{Q}_{\zeta^D}(w)\big] \\
&= \mathbb{E}_{\zeta = (\zeta^D, \zeta^T, \zeta^R)}\big[\sum_{w \in \mathcal{W}} \boldsymbol{Q}_{\zeta^D}(w) \cdot \mathbf{1}_{G(\zeta^R) < \min\{1, \boldsymbol{P}(w)/\boldsymbol{Q}(w)\}}\big] \\
&= \sum_{w \in \mathcal{W}} \mathbb{E}_{\zeta^D}[\boldsymbol{Q}_{\zeta^D}] \cdot \mathbb{E}_{\zeta^R}[\mathbf{1}_{G(\zeta^R) < \min\{1, \boldsymbol{P}(w)/\boldsymbol{Q}(w)\}}] \\
&= \sum_{w \in \mathcal{W}} \boldsymbol{Q}(w) \cdot \min\left\{1, \frac{\boldsymbol{P}(w)}{\boldsymbol{Q}(w)}\right\} = 1 - \mathrm{TV}(\boldsymbol{Q}, \boldsymbol{P}),
\end{aligned}
$$

where independence of $\zeta^D$ and $\zeta^R$ is used in the third equality. Thus, Alg. 1 can preserve the sampling efficiency.

Finally, we show (c). Recall from the proof of (a) that there are two cases. **Case 1**: The expression of $\boldsymbol{P}'_\zeta$ is given by 15. Since $\mathcal{S}$ achieves the largest watermark strength, we have $\boldsymbol{Q}_{\zeta^D} = \mathcal{S}(\boldsymbol{Q}, \zeta^D)$ as a degenerate distribution. It then follows that

$$
\boldsymbol{P}'_\zeta(w) = \boldsymbol{Q}_{\zeta^D}(w) \cdot \mathbf{1}_{G(\zeta^R) < \min\{1, \frac{\boldsymbol{P}(w')}{\boldsymbol{Q}(w')}\}} + (\boldsymbol{P} - \boldsymbol{Q})_{+, \zeta^T}(w) \cdot \mathbf{1}_{G(\zeta^R) \geq \min\{1, \frac{\boldsymbol{P}(w')}{\boldsymbol{Q}(w')}\}},
$$

where $w'$ is the token sampled by $\boldsymbol{Q}_{\zeta^D}$. Moreover, $(\boldsymbol{P} - \boldsymbol{Q})+, \zeta^T$ is also degenerate, which implies that $\boldsymbol{P}'\zeta$ is always degenerate and that $\mathsf{WS}(\boldsymbol{P}'_\zeta) = \mathrm{Ent}(\boldsymbol{P})$. **Case 2**: $\boldsymbol{P}_{\zeta^T}$ is itself degenerate.

Therefore, in both cases, the final distribution produced by Alg. 1 is always degenerate, and hence the watermark strength is preserved. $\qquad\square$

## E  BAYESIAN SCORING FUNCTIONS FOR SYNTHID

Here we introduce the details of the Bayesian scoring function mentioned in Section 4.2. For a given text, we have two hypotheses: unwatermarked ($H_0$) and watermarked ($H_1$). Our target is to estimate the posterior $\mathbb{P}(H_1|y^D, y^T, u)$, which is the probability that the text is watermarked given its $g$-values and acceptance variable. Formally,

$$
\begin{aligned}
\mathbb{P}(H_1 \mid y^D, y^T, u) = \ \sigma\Big(&\log \mathbb{P}(y^D, y^T \mid H_1, u) - \log \mathbb{P}(y^D, y^T \mid H_0, u) \\
&+ \log \mathbb{P}(H_1 \mid u) - \log \mathbb{P}(H_0 \mid u)\Big)
\end{aligned}
\tag{16}
$$

where $\sigma(\cdot)$ is the sigmoid function. There are two terms that need to be estimated in equation 16. First, the prior $\mathbb{P}(H_1 \mid u)$, which can be learned empirically. Actually, $u$ does not influence the existence of the watermark, so $\mathbb{P}(H_1 \mid u) = \mathbb{P}(H_1)$ (also $\mathbb{P}(H_0 \mid u) = \mathbb{P}(H_0)$), which is the prior probability that a text is watermarked (or not). Following the setting in Dathathri et al. (2024), we set this prior to 0.5 in our experiments. Second, the likelihood $\mathbb{P}(y^D, y^T \mid H_1, u)$ and $\mathbb{P}(y^D, y^T \mid H_0, u)$. To illustrate, we fix step $t$ and tournament layer $l$. Due to the independence across tokens and layers, once the odds are determined for a fixed step and layer, they can be multiplied to obtain the odds for the entire sequence. Recall that for step $t$, we denote $y_t^D = (\boldsymbol{g}_{t,1}^D(w_t), \ldots, \boldsymbol{g}_{t,m}^D(w_t)) \in \mathbb{R}^m$ with a corresponding definition for $y_t^T$. For notational simplicity, we write $g_{t,l}^D = \boldsymbol{g}_{t,1}^D(w_t)$. The unwatermarked likelihood is then given by

$$
\mathbb{P}(g_{t,l}^D, g_{t,l}^T \mid H_0, u_t) = f_g(g_{t,l}^D)f_g(g_{t,l}^T),
$$

where $f_g$ denotes the probability mass function of the $g$-value. In our setting, $g$-values follow Bernoulli$(0.5)$, so $f_g = 0.5$. For the watermarked likelihoods, we have:

$$\mathbb{P}(g_{t,l}^D, g_{t,l}^T \mid H_1, u_t) = \sum_{\kappa \in \{D,T\}} \mathbb{P}(g_{t,l}^D, g_{t,l}^T \mid \zeta_t = \zeta_t^\kappa, H_1, u_t)\, \mathbb{P}(\zeta_t = \zeta_t^\kappa \mid H_1, u_t)$$

$$= \mathbb{P}(g_{t,l}^D \mid \zeta_t = \zeta_t^D)\, f_g(g_{t,l}^T)\, \mathbb{P}(\zeta_t = \zeta_t^D \mid u_t) \qquad (17)$$

$$+ \mathbb{P}(g_{t,l}^T \mid \zeta_t = \zeta_t^T)\, f_g(g_{t,l}^D)\left(1 - \mathbb{P}(\zeta_t = \zeta_t^D \mid u_t)\right),$$

where $\zeta_t = \zeta_t^D$ indicates that the token at step $t$ was watermarked with pseudorandom seed $\zeta^D$. Two terms in equation 17 require estimation. First, $\mathbb{P}(\zeta_t = \zeta_t^D \mid u_t)$, which means given the acceptance variable, the probability that the token comes from the draft. Second, $\mathbb{P}(g_{t,l}^\kappa \mid \zeta_t = \zeta_t^\kappa)$, which is the likelihood of the g-values under a specific pseudorandom seed. Following Dathathri et al. (2024), when SynthID employs two-sample tournament sampling, we can factorize $\mathbb{P}(g_{t,l}^\kappa \mid \zeta_t = \zeta_t^\kappa)$ and then have,

$$\mathbb{P}(g_{t,l}^D, g_{t,l}^T \mid H_1, u_t) = \mathbb{P}(g_{t,l}^D \mid \zeta_t = \zeta_t^D)\, f_g(g_{t,l}^T)\, \mathbb{P}(\zeta_t = \zeta_t^D \mid u_t)$$

$$+ \mathbb{P}(g_{t,l}^T \mid \zeta_t = \zeta_t^T)\, f_g(g_{t,l}^D)\left(1 - \mathbb{P}(\zeta_t = \zeta_t^D \mid u_t)\right)$$

$$= \frac{1}{4}\left[(g_{t,l}^D - \tfrac{1}{2})\,\mathbb{P}(\psi_{t,l}^D = 2 \mid g_{t,<l}^D) + 1\right]\mathbb{P}(\zeta_t = \zeta_t^D \mid u_t)$$

$$+ \frac{1}{4}\left[(g_{t,l}^T - \tfrac{1}{2})\,\mathbb{P}(\psi_{t,l}^T = 2 \mid g_{t,<l}^T) + 1\right]\left(1 - \mathbb{P}(\zeta_t = \zeta_t^D \mid u_t)\right),$$

where $\psi_{t,l}$ is a random variable denoting the number of unique tokens appearing in the tournament match at layer $l$ and step $t$. We model $\mathbb{P}(\psi_{t,l}^\kappa = 2 \mid g_{t,<l}^\kappa)$ using logistic regression:

$$\mathbb{P}\left(\psi_{t,l}^\kappa = 2 \mid g_{t,<l}^\kappa\right) = \sigma\left(\beta_l^\kappa + \sum_{j=1}^{l-1} \delta_{l,j}^\kappa g_{t,j}^\kappa\right),$$

where $\sigma(\cdot)$ is the sigmoid function. Here, $\beta_l^\kappa \in \mathbb{R}$ is the bias term for layer $l$, and $\delta_{l,j}^\kappa \in \mathbb{R}$ represents the influence of $g_{t,j}^\kappa$ on the probability that $\psi_{t,l}^\kappa = 2$. The remaining term to estimate is $\mathbb{P}(\zeta_t = \zeta_t^D \mid u_t)$. Without access to the acceptance variable, Dathathri et al. (2024) treats this probability as a prior, estimated directly from the acceptance rate of speculative sampling; we refer to this approach as **Bayes-Prior**. In our method, we leverage the acceptance variable and train a three-layer MLP to perform the estimation:

$$\mathbb{P}(\zeta_t = \zeta_t^D \mid u_t) = \begin{cases} \sigma\left(\alpha(\tau_t - u_t)\right), & \text{for training,} \\ \mathbf{1}_{u_t \leq \tau_t}, & \text{for inference,} \end{cases}$$

where $\sigma(\cdot)$ is the sigmoid function, $\alpha$ is a scaling parameter, and $\tau_t = \mathrm{MLP}(x_t)$ with input $x_t = [g_{t,1}^D, \ldots, g_{t,30}^D, g_{t,1}^T, \ldots, g_{t,30}^T] \in \mathbb{R}^{60}$. We denote our method as **Bayes-MLP**.

In summary, **Bayes-Prior** relies solely on the $g$-values for detection:

$$\textbf{Bayes-Prior}(y^D, y^T) = \mathbb{P}(H_1 \mid y^D, y^T),$$

whereas **Bayes-MLP** additionally incorporates the pseudorandom variable $u$:

$$\textbf{Bayes-MLP}(y^D, y^T, u) = \mathbb{P}(H_1 \mid y^D, y^T, u),$$

with $y^D, y^T \in \mathbb{R}^{m \times N}$ and $u \in \mathbb{R}^N$, where $N$ denotes the token sequence length.

# F EXPERIMENT DETAILS AND RESULTS

## F.1 IMPLEMENTATION DETAILS

**Implementation of Ars-$\tau$.** In Section 4.2, we introduced the use of the pseudorandom variable $u$ and proposed **Ars-$\tau$** for the Gumbel-max watermark. Recall that the selection rule for **Ars-$\tau$** is defined as

$$y_t = y_t^D \mathbf{1}_{G(\zeta_t^R) < \tau} + y_t^T \mathbf{1}_{G(\zeta_t^R) \geq \tau}.$$

where $\tau \in [0, 1]$. To determine the optimal $\tau$, we adopt a straightforward approach: on the training set, we evaluate 100 evenly spaced values of $\tau$ over $[0, 1]$ and select the one that performs best, which is then applied to the test set.

**Implementation of Bayes-MLP.**    Details are provided in Appendix E.

## F.2    Additional Experimental Results

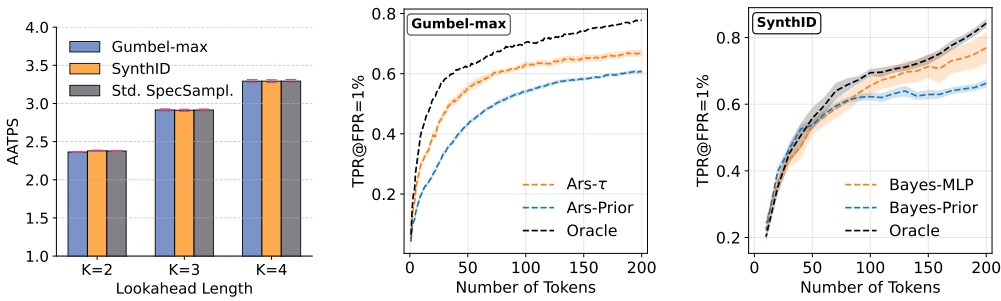

Figure 3: Experimental results for `Gemma` models on the `ELI5` dataset. **Left**: Average Accepted Tokens Per Step (AATPS) of Alg. 1 applied to the Gumbel-max and SynthID watermarks, compared with Standard Speculative Sampling (Std. SpecSampl). Error bars mark the 95% confidence intervals. **Middle** and **Right**: Watermark detectability (TPR at FPR = 1%) for Alg. 1 on the Gumbel-max (middle) and SynthID (right). Orange curves show our method, blue curves show the prior-based method, and black curves represent the ideal detector (Oracle) that always selects the correct test statistic. Shaded regions indicate the 95% confidence intervals.

Table 1: Results of Alg. 1 applied to the Gumbel-max and SynthID watermarks on the `ELI5` dataset, compared with Standard Speculative Sampling (Std. SpecSampl.) and basic watermarks. **AATPS**: Average Accepted Tokens Per Step; **PTT**: Per Token Time in millisecond; **LOGPPL**: Log Perplexity.

| Models | Lookahead | Method | AATPS | PTT | LOGPPL |
|--------|-----------|--------|-------|-----|--------|
| Llama-7b / Llama-68m | basic | Gumbel-max | $1.0 \pm 0.0$ | $22.09 \pm 0.151$ | $2.08 \pm 0.024$ |
| | | SynthID | $1.0 \pm 0.0$ | $44.94 \pm 0.477$ | $2.10 \pm 0.017$ |
| | $K = 2$ | Gumbel-max | $1.7707 \pm 0.0058$ | $17.04 \pm 0.121$ | $2.16 \pm 0.025$ |
| | | SynthID | $1.7645 \pm 0.0042$ | $37.15 \pm 0.419$ | $2.12 \pm 0.014$ |
| | | Std. SpecSampl. | $1.7666 \pm 0.0099$ | $14.98 \pm 0.167$ | $2.18 \pm 0.023$ |
| | $K = 3$ | Gumbel-max | $1.9650 \pm 0.0082$ | $17.00 \pm 0.131$ | $2.14 \pm 0.024$ |
| | | SynthID | $1.9584 \pm 0.0066$ | $40.75 \pm 0.554$ | $2.13 \pm 0.015$ |
| | | Std. SpecSampl. | $1.9577 \pm 0.0070$ | $15.77 \pm 0.059$ | $2.07 \pm 0.022$ |
| | $K = 4$ | Gumbel-max | $2.0987 \pm 0.0095$ | $17.96 \pm 0.141$ | $2.16 \pm 0.024$ |
| | | SynthID | $2.0927 \pm 0.0078$ | $41.74 \pm 0.658$ | $2.15 \pm 0.014$ |
| | | Std. SpecSampl. | $2.0988 \pm 0.0094$ | $15.56 \pm 0.074$ | $2.19 \pm 0.022$ |
| Gemma-7b / Gemma-2b | basic | Gumbel-max | $1.0 \pm 0.0$ | $29.20 \pm 0.072$ | $1.79 \pm 0.053$ |
| | | SynthID | $1.0 \pm 0.0$ | $41.32 \pm 0.179$ | $1.69 \pm 0.037$ |
| | $K = 2$ | Gumbel-max | $2.3637 \pm 0.0085$ | $25.98 \pm 0.175$ | $1.69 \pm 0.055$ |
| | | SynthID | $2.3794 \pm 0.0089$ | $38.10 \pm 0.522$ | $1.74 \pm 0.036$ |
| | | Std. SpecSampl. | $2.3773 \pm 0.0080$ | $22.74 \pm 0.082$ | $1.66 \pm 0.057$ |
| | $K = 3$ | Gumbel-max | $2.9146 \pm 0.0144$ | $26.83 \pm 0.204$ | $1.75 \pm 0.053$ |
| | | SynthID | $2.9108 \pm 0.0142$ | $35.62 \pm 0.291$ | $1.75 \pm 0.035$ |
| | | Std. SpecSampl. | $2.9140 \pm 0.0140$ | $23.84 \pm 0.121$ | $1.69 \pm 0.051$ |
| | $K = 4$ | Gumbel-max | $3.2923 \pm 0.0203$ | $28.94 \pm 0.245$ | $1.75 \pm 0.052$ |
| | | SynthID | $3.2920 \pm 0.0190$ | $41.06 \pm 0.444$ | $1.73 \pm 0.037$ |
| | | Std. SpecSampl. | $3.2930 \pm 0.0213$ | $26.05 \pm 0.179$ | $1.72 \pm 0.053$ |

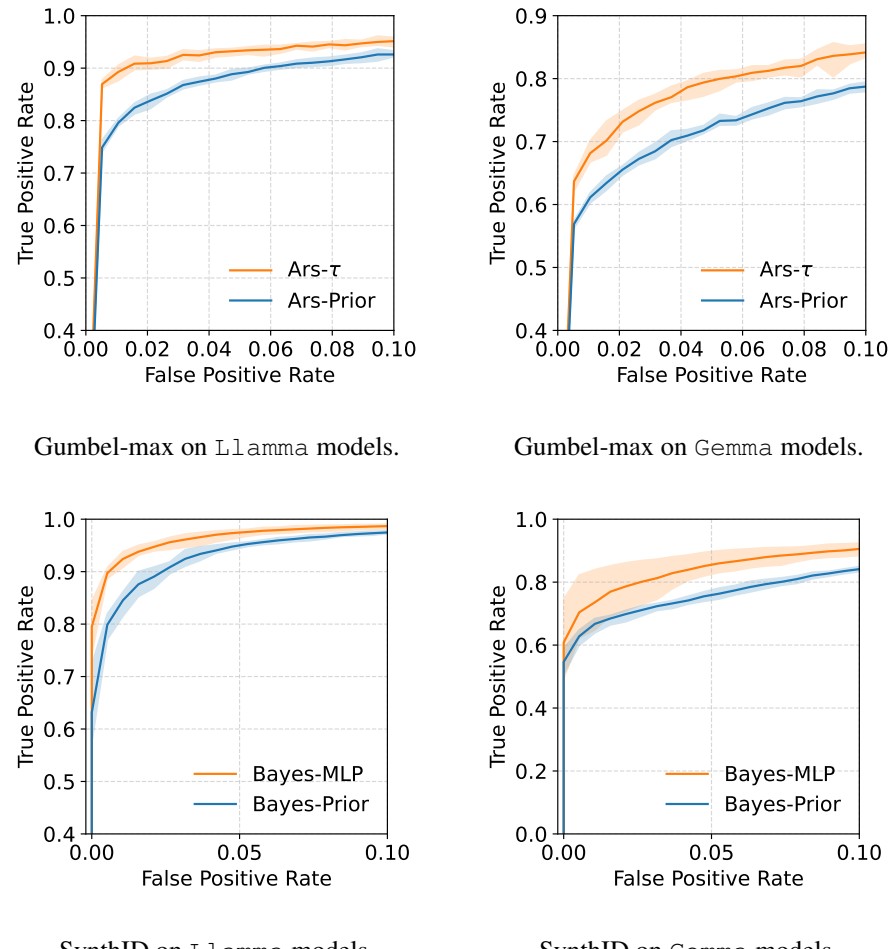

Gumbel-max on `Llamma` models.

Gumbel-max on `Gemma` models.

SynthID on `Llamma` models.

SynthID on `Gemma` models.

Figure 4: ROC curves for watermark detection on the `ELI5` dataset. Gumbel-max performance is evaluated at a token length of 200, while SynthID performance is evaluated at a token length of 100. Orange curves show our method, and blue curves show the prior-based method.

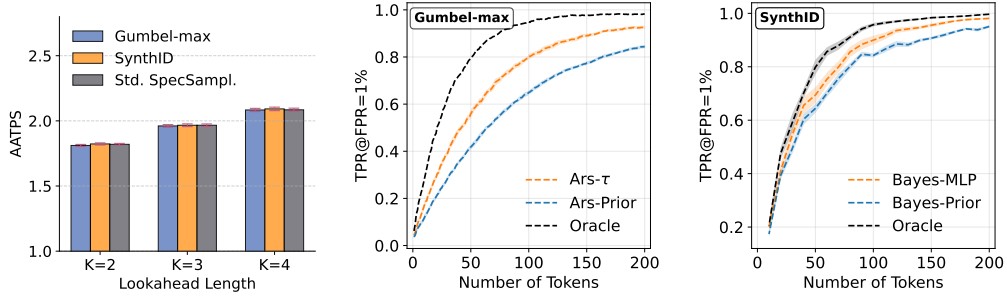

Figure 5: Experimental results for `Llama` models on the `C4` dataset. **Left**: Average Accepted Tokens Per Step (AATPS) of Alg. 1 applied to the Gumbel-max and SynthID watermarks, compared with Standard Speculative Sampling (Std. SpecSampl). Error bars mark the 95% confidence intervals. **Middle** and **Right**: Watermark detectability (TPR at FPR = 1%) for Alg. 1 on the Gumbel-max (middle) and SynthID (right). Orange curves show our method, blue curves show the prior-based method, and black curves represent the ideal detector (Oracle) that always selects the correct test statistic. Shaded regions indicate the 95% confidence intervals.

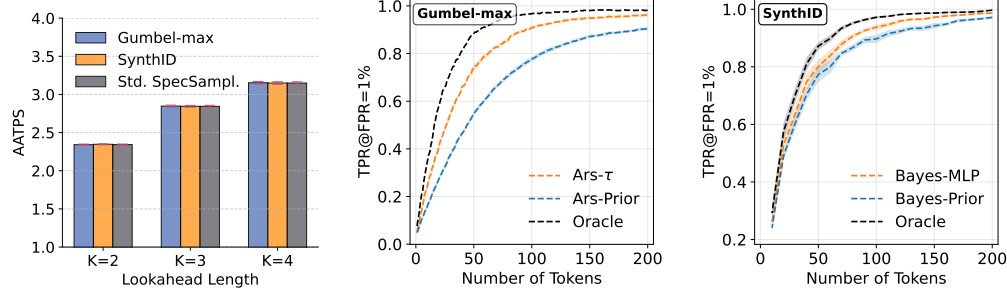

Figure 6: Experimental results for `Gemma` models on the `C4` dataset. **Left**: Average Accepted Tokens Per Step (AATPS) of Alg. 1 applied to the Gumbel-max and SynthID watermarks, compared with Standard Speculative Sampling (Std. SpecSampl). Error bars mark the 95% confidence intervals. **Middle** and **Right**: Watermark detectability (TPR at FPR = 1%) for Alg. 1 on the Gumbel-max (middle) and SynthID (right). Orange curves show our method, blue curves show the prior-based method, and black curves represent the ideal detector (Oracle) that always selects the correct test statistic. Shaded regions indicate the 95% confidence intervals.

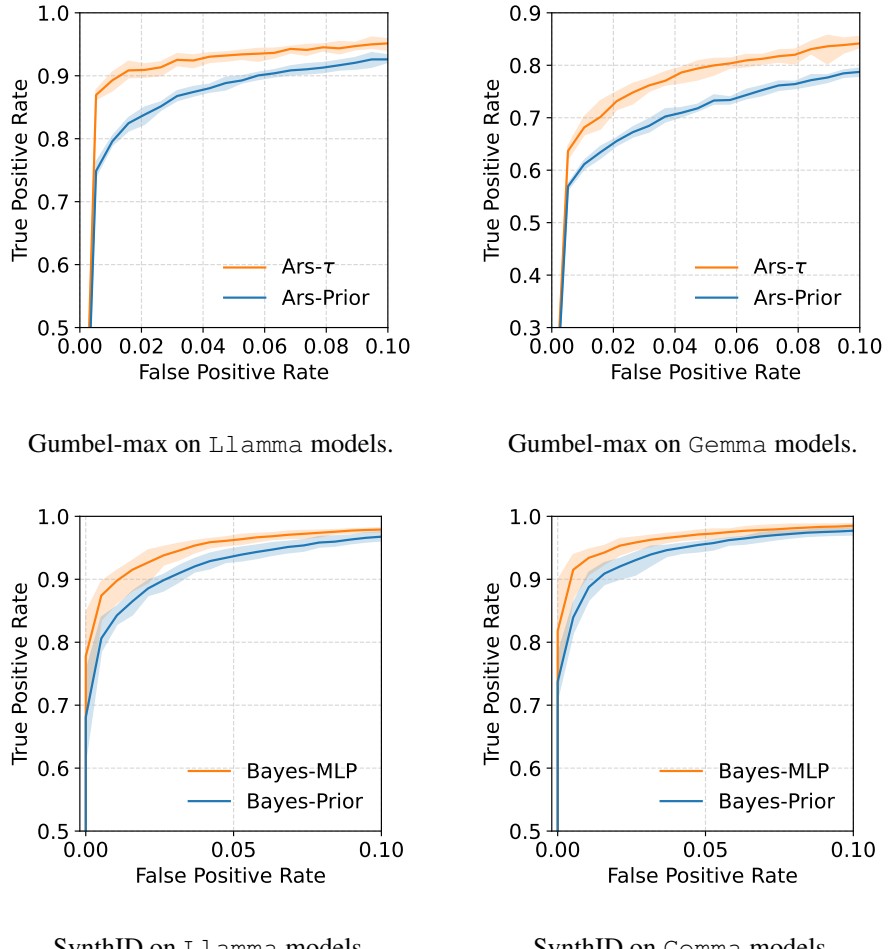

Gumbel-max on `Llamma` models.

Gumbel-max on `Gemma` models.

SynthID on `Llamma` models.

SynthID on `Gemma` models.

Figure 7: ROC curves for watermark detection on the `C4` dataset. Gumbel-max performance is evaluated at a token length of 200, while SynthID performance is evaluated at a token length of 100. Orange curves show our method, and blue curves show the prior-based method.

Table 2: Results of Alg. 1 applied to the Gumbel-max and SynthID watermarks on the `C4` dataset, compared with Standard Speculative Sampling (Std. SpecSampl.) and basic watermarks. **AATPS**: Average Accepted Tokens Per Step; **PTT**: Per Token Time in millisecond; **LOGPPL**: Log Perplexity.

| Models | Lookahead | Method | AATPS | PTT | LOGPPL |
|---|---|---|---|---|---|
| `Llama-7b` / `Llama-68m` | basic | Gumbel-max | $1.0 \pm 0.0$ | $22.98 \pm 0.040$ | $2.22 \pm 0.023$ |
| | | SynthID | $1.0 \pm 0.0$ | $40.27 \pm 0.307$ | $2.29 \pm 0.015$ |
| | $K = 2$ | Gumbel-max | $1.8112 \pm 0.0080$ | $17.65 \pm 0.062$ | $2.21 \pm 0.023$ |
| | | SynthID | $1.8228 \pm 0.0081$ | $35.44 \pm 0.404$ | $2.24 \pm 0.016$ |
| | | Std. SpecSampl. | $1.8200 \pm 0.0071$ | $16.01 \pm 0.044$ | $2.28 \pm 0.023$ |
| | $K = 3$ | Gumbel-max | $1.9610 \pm 0.0099$ | $17.70 \pm 0.070$ | $2.22 \pm 0.025$ |
| | | SynthID | $1.9662 \pm 0.0106$ | $39.62 \pm 0.552$ | $2.25 \pm 0.015$ |
| | | Std. SpecSampl. | $1.9663 \pm 0.0101$ | $15.69 \pm 0.056$ | $2.26 \pm 0.022$ |
| | $K = 4$ | Gumbel-max | $2.0836 \pm 0.0125$ | $18.21 \pm 0.081$ | $2.22 \pm 0.024$ |
| | | SynthID | $2.0917 \pm 0.0135$ | $39.48 \pm 0.672$ | $2.24 \pm 0.015$ |
| | | Std. SpecSampl. | $2.0847 \pm 0.0128$ | $15.83 \pm 0.068$ | $2.23 \pm 0.023$ |
| `Gemma-7b` / `Gemma-2b` | basic | Gumbel-max | $1.0 \pm 0.0$ | $29.19 \pm 0.006$ | $2.52 \pm 0.026$ |
| | | SynthID | $1.0 \pm 0.0$ | $37.92 \pm 0.193$ | $2.56 \pm 0.013$ |
| | $K = 2$ | Gumbel-max | $2.3415 \pm 0.0077$ | $25.67 \pm 0.061$ | $2.53 \pm 0.026$ |
| | | SynthID | $2.3468 \pm 0.0070$ | $35.61 \pm 0.396$ | $2.58 \pm 0.014$ |
| | | Std. SpecSampl. | $2.3427 \pm 0.0074$ | $23.57 \pm 0.054$ | $2.60 \pm 0.027$ |
| | $K = 3$ | Gumbel-max | $2.8473 \pm 0.0127$ | $28.19 \pm 0.110$ | $2.53 \pm 0.026$ |
| | | SynthID | $2.8450 \pm 0.0119$ | $33.14 \pm 0.441$ | $2.59 \pm 0.014$ |
| | | Std. SpecSampl. | $2.8442 \pm 0.0129$ | $2.50 \pm 0.026$ | $2.50 \pm 0.026$ |
| | $K = 4$ | Gumbel-max | $3.1529 \pm 0.0176$ | $28.98 \pm 0.145$ | $2.53 \pm 0.027$ |
| | | SynthID | $3.1499 \pm 0.0165$ | $36.84 \pm 0.472$ | $2.59 \pm 0.015$ |
| | | Std. SpecSampl. | $3.1494 \pm 0.0164$ | $26.61 \pm 0.100$ | $2.60 \pm 0.027$ |

## F.3 THEORETICAL SPEEDUP VS. EMPIRICAL RUNTIMES

Both Average Accepted Tokens Per Step (AATPS) and Per Token Time (PTT) reported in Table 1, 2 reflect how much Alg. 1 accelerates the generation process, but they capture different aspects of performance. AATPS measures the number of accepted tokens in each generation loop and thus reflects the theoretical speedup, focusing primarily on the draft token acceptance rate, to which Alg. 1 directly contributes. Ideally, a higher acceptance rate leads to a greater overall speedup. In contrast, PTT empirically measures the actual runtime of generation and can be affected by various factors, including (but not limited to) watermark sampling, token verification, and model switching (when using a single GPU). Therefore, the observed speedup based on PTT may not perfectly align with that implied by AATPS. We do not further investigate this discrepancy, as it falls beyond the scope of this work.

## G THE USE OF LARGE LANGUAGE MODELS (LLMS)

The LLMs were used solely to polish the writing, including grammar correction and improvements in clarity and style. The research contributions, including the design of methods, implementation, experiments, and analysis, were carried out entirely by the authors.

