# OpenReview forum: "Improving the Trade-off Between Watermark Strength and Speculative Sampling Efficiency for Language Models"
_ICLR.cc/2026/Conference — ICLR 2026 Poster_

### Official Review · Reviewer_Zd3w · 2025-10-29

**Soundness:** 3
**Presentation:** 3
**Contribution:** 3
**Rating:** 6
**Confidence:** 4

**Summary:**

This paper revisits the trade-off between watermark strength and speculative sampling efficiency in large language models (LLMs). The authors argue that the incompatibility stems from the use of independent random sources: watermarking modifies token sampling using pseudorandomness, while speculative sampling introduces additional stochasticity through draft-token acceptance. This mismatch weakens watermark detectability and limits inference efficiency. To resolve this, the paper proposes a unified pseudorandom framework that decomposes random sources into three independent components: \(ζ_D\) (for the draft model), \(ζ_T\) (for the target model), and \(ζ_R\) (for acceptance decisions)

**Strengths:**

- The paper provides a clear theoretical explanation for why speculative sampling and watermarking interfere with each other—namely, their reliance on separate random sources.
- The proposed pseudorandom acceptance mechanism is simple yet elegant, turning the generation process into a deterministic function of pseudorandom variables.
- Theoretical analysis (Pareto frontier characterization) and empirical evidence consistently demonstrate that the proposed design breaks the previously assumed trade-off between watermark strength and efficiency.
- The work unifies two practically important but previously conflicting mechanisms, offering a potential path toward efficient and traceable LLM inference

**Weaknesses:**

1. The experimental evaluation only considers two unbiased watermarking schemes (Gumbel-max and SynthID). The analysis would be more convincing if extended to other representative watermarking frameworks, such as KGW or reweight-based methods [1,2].

2. The proposed method requires applying watermarking to both the draft and target models during speculative sampling. The paper does not analyze whether this dual watermarking might affect text quality or fluency.

[1] Hu, Zhengmian, Lichang Chen, Xidong Wu, Yihan Wu, Hongyang Zhang, and Heng Huang. *Unbiased watermark for large language models.* arXiv preprint arXiv:2310.10669 (2023).
[2] Wu, Y., Hu, Z., Guo, J., Zhang, H., & Huang, H. *A resilient and accessible distribution-preserving watermark for large language models.* arXiv preprint arXiv:2310.07710 (2023).

**Questions:**

1. Could the authors discuss whether the pseudorandom control framework can be generalized to biased or reweighted watermark schemes?

2. Considering that speculative sampling is primarily designed to accelerate inference, the paper should report additional runtime measurements comparing (a) standard speculative sampling, (b) speculative sampling with watermarking, and (c) the proposed method, to confirm that efficiency gains are retained.

---

> ### Author Response · Authors · 2025-11-21
> **Response to Reviewer**
>
> Thanks for your professional review and positive feedback! The following are our responses:
>
> ### **1. Extend to other unbiased watermark**
>
> Thanks for raising this point. We evaluate our method using two representative unbiased watermarks—**Gumbel-max** (OpenAI) and **SynthID** (Google). Our focus is on *characterizing and improving the trade-off between watermark strength and sampling efficiency* rather than exhaustive empirical benchmarking for various watermarks. We believe the current experiments are sufficient to support our main contributions.
>
> Additionally, the proposed method can *not* be applied to KGW. It can be directly applied to **any unbiased watermark** that already achieves **maximum watermark strength**, which requires the watermarked distribution to be **degenerate**—i.e., all probability mass concentrated on a single token. Notice that the KGW is *not* unbiased and does *not* attain maximal watermark strength. Also, $\gamma$ Reweigh in [1] and DiPmark in [2] are **unbiased but non-maximum-strength** schemes. While our framework can be applied to the latter two, it cannot elevate their watermark strength to the maximum, since their decoders are intrinsically non-deterministic.
>
> More discussion about generalization to other watermarks is discussed under Question 1.
>
> ---
>
> ### **2. Analyze the influence on text quality or fluency**
>
> Thanks for this great suggestion. We now report **log-perplexity** in the updated *Table 1*, following the reviewer’s suggestion. The results show that our method—including pseudorandom acceptance—does **not degrade text quality** compared to standard speculative sampling. More broadly, text quality can be influenced by multiple components (speculative sampling strategy, watermarking scheme, draft–target model choice, etc.), and a deeper study is beyond the scope of this work. We have also listed text-quality impacts as part of the future work (see *line 485*).
>
> ---
>
> ### Responses to Questions:
>
> 1. **Generalization to biased or reweighted watermark schemes**. We make the following clarifications.
>
>    - **Unbiased watermark is necessary for our framework**. The notion of maximum strength is defined **under the assumption of unbiasedness**, which ensures that watermarking does not alter the original token distribution $P$. Therefore, **biased** watermark schemes cannot be directly incorporated into our framework. We also note that unbiased watermarks are generally more desirable in practice because they avoid the distribution distortion.
>
>    - For **unbiased reweighted watermark schemes** that *do not* achieve maximum watermark strength—such as $\gamma$ Reweigh in [1] and DiPmark in [2]—our framework can still be applied. However, it **cannot elevate** these schemes to maximum strength, since the watermark itself remains non-deterministic. The pseudorandom acceptance mechanism removes randomness from speculative sampling, but it cannot remove intrinsic randomness in the watermarking decoder. Exploring how to improve the watermark strength of such non-maximum unbiased schemes is an interesting direction for future work.
>
> 2. **Empirical runtime**. Thanks for this suggestion. We now report **Per Token Time (PTT)** in the updated *Table 1*, following the reviewer’s suggestion. The results show that, compared with basic watermarking, our method (watermarking with speculative sampling and pseudorandom acceptance) does **improve sampling speed**.
>
>    We also want to clarify the distinction between **theoretical speedup** and **empirical runtime**. In Table 1, Average Accepted Tokens Per Step (AATPS) measures the number of accepted tokens in each generation loop and thus reflects the theoretical speedup, focusing primarily on the draft token acceptance rate, to which our method **directly contributes**. In contrast, PTT empirically measures the actual runtime of generation and can be affected by *various factors*, including (but not limited to) watermark sampling, token verification, and model switching (when using a single GPU). Therefore, the observed speedup based on PTT may not perfectly align with that implied by AATPS, and further runtime analysis lies outside the scope of our main contributions.

---

### Official Review · Reviewer_sUKk · 2025-10-31

**Soundness:** 3
**Presentation:** 3
**Contribution:** 3
**Rating:** 6
**Confidence:** 3

**Summary:**

The paper revisits the trade-off between watermark strength and speculative sampling efficiency in large language models. It formally quantifies watermark strength using KL divergence and derives a theoretical upper bound equal to the entropy of the base model. To overcome the claimed inevitable trade-off from prior work, the authors propose a pseudorandom acceptance mechanism that makes the entire sampling process a deterministic function of pseudorandom seeds. They further introduce new detection methods (Ars-$\tau$ and Bayes-MLP) leveraging this mechanism, and demonstrate through experiments that the approach improves detection performance without reducing decoding throughput.

**Strengths:**

The paper provides a theoretical framework connecting watermark strength and speculative-sampling efficiency. The proposed pseudorandom-acceptance mechanism is theoretically appealing, showing that the trade-off can be alleviated under certain conditions.

**Weaknesses:**

1. The proof that watermark strength can reach the upper bound $WS = H(P)$ (Theorem 3.2) assumes that each $P_\zeta$ is a completely deterministic distribution, where every generated token is fixed once the pseudorandom seed is given. However, even SynthID’s tournament sampling still has randomness after top-$k$ and $2^m$ rounds, so $P_\zeta$ isn’t truly deterministic. This means the bound is more of a theoretical ideal than something achievable in real LLM decoding. The paper should make this distinction clearer and discuss how far practical schemes are from that ideal. For example, how much does it improve over different m (the number of layers used in SynthID)?

2. The claim of achieving both the best sampling efficiency ($SSE = 1 - \mathrm{TV}(Q, P)$) and the highest watermark strength ($WS = H(P)$) relies on several strong assumptions, such as perfectly unbiased decoders, fully deterministic generation, and independent pseudorandom seeds. If any of these assumptions don’t hold, the trade-off comes back. So should the result really be interpreted as showing what’s possible under idealized conditions, not as a general solution that always works?

**Questions:**

see weaknesses

---

> ### Author Response · Authors · 2025-11-21
> **Response to Reviewer**
>
> Thanks for your professional review and positive feedback! The following are our responses:
>
> ### **1. Randomness for SynthID**
>
> Thanks for raising this question. We make the following clarifications.
>
> - First, the upper bound in Theorem 3.2 is **achievable in real LLM decoding**. Note that the Gumbel-max watermark—one of the most widely used unbiased watermarking schemes—*does* achieve the maximal watermark strength in practice. Its decoder is fully deterministic: once the pseudorandom seed is given, the selected token is fixed.
>
> - Second, for SynthID, the tournament sampling is stochastic when the number of rounds $m$ is finite. However, as shown in Appendix B.3, when $m \to \infty$, the sampling distribution **converges almost surely** to a degenerate distribution. In this limit, the output token becomes a deterministic function of the pseudorandom variable, and SynthID attains maximal watermark strength in exactly the sense described in Theorem 3.2. To clearly distinguish the theoretical infinite-round case from practical finite-round implementations, we **explicitly compare them in Section 3.2** (see *Figure 1*). The figure shows that while $m=\infty$ achieves the theoretical maximum (blue curves), the practical setting $m=30$ yields a noticeably **reduced watermark strength** (gray curve), consistent with our theoretical analysis.
>
> ----
>
> ### **2. Strong assumptions**
>
> Thanks for raising this concern. We clarify here that these assumptions are *not* idealized or unrealistic—they are **standard in the unbiased watermarking literature** [1-5] and are **achievable in practical LLM decoding**. Specifically:
>
> - Unbiased decoders. The decoder’s unbiasedness is ensured by the underlying watermarking strategy. For example, the Gumbel-max watermark rigorously satisfies the unbiasedness property through the Gumbel-max trick, which guarantees that the expected watermarked distribution matches the original next-token distribution.
> - Fully deterministic generation. As we discussed above,  fully deterministic generation is **achievable** in real LLM decoding (e.g., Gumbel-max watermark).
> - Independent pseudorandom seeds. Independence among pseudorandom seeds is **straightforward in practice**, as modern PRNGs (such as Philox (used in JAX/PyTorch) or Mersenne Twister) are explicitly designed to provide independent draws at negligible cost. Moreover, we apply repeated-context masking to avoid reusing the same pseudorandom seed, which further preserves independence in implementation.
>
> Unbiased decoders and fully deterministic generation are both achievable in practice. That said, we acknowledge that ideal independent pseudorandom seeds assumptions may not hold perfectly in all practical deployments. However, our empirical results for Log Perplexity (see Table 1) indicate that **minor (possible) deviations from independence have a negligible impact on the unbiasedness property** (the independence assumption ensures the unbiasedness property for Algorithm 1). This suggests that our method delivers the intended trade-off improvements under realistic and general conditions.
>
> ---
>
> ### **References**
>
> [1] Li, Xiang, et al. "Robust detection of watermarks for large language models under human edits." *Journal of the Royal Statistical Society Series B: Statistical Methodology* (2025): qkaf056.
>
> [2] Kuditipudi, Rohith, et al. "Robust distortion-free watermarks for language models." arXiv preprint arXiv:2307.15593 (2023).
>
> [3] Li, Xiang, et al. "A statistical framework of watermarks for large language models: Pivot, detection efficiency and optimal rules." The Annals of Statistics 53.1 (2025): 322-351.
>
> [4] Hu, Zhengmian, et al. "Unbiased watermark for large language models." *arXiv preprint arXiv:2310.10669* (2023).
>
> [5] Dathathri, Sumanth, et al. "Scalable watermarking for identifying large language model outputs." *Nature* 634.8035 (2024): 818-823.

---

### Official Review · Reviewer_d8rs · 2025-11-01

**Soundness:** 4
**Presentation:** 4
**Contribution:** 3
**Rating:** 8
**Confidence:** 4

**Summary:**

The paper revisits the trade-off between watermarking and speculative sampling. It (i) quantifies watermark strength as the expected KL divergence, showing this metric controls the asymptotic p‑value decay rate of the optimal detector (ii) formalizes the strength–efficiency Pareto frontier (iii) proposes pseudorandom acceptance by introducing an extra seed $\zeta_R$ so the accept/reject step is itself a deterministic function of pseudorandomness. Empirically, the method matches standard speculative sampling in AATPS while improving TPR@FPR=1% for detection compared with previous methods that preserve AATPS.

**Strengths:**

1. New perspective (not detection‑first, measure strength via KL).
The paper propose to cleanly separates watermark strength from detector performance, grounding the former in an information‑theoretic quantity (expected KL) with a clear sample‑complexity interpretation (Theorem 3.1) and a tight entropy upper bound (Theorem 3.2). This presents a principled framework for further analysis.

2. Evidence of better trade‑offs in practice.
AATPS closely tracks the standard baseline, while detection improves compared with previous methods that preserve AATPS.

3. Clear writing and organization.
The paper is easy to follow, with crisp statements, clear algorithm, and figures that directly support claims.

**Weaknesses:**

### Whether the “trade‑off” truly disappears (terminology/interpretation).

Theorem 4.1. builds on redefining strength as expected KL. This metrics makes some new proposed analysis possible. However, the paper itself clarifies that watermark strength ≠ detection efficiency (Remark 3.1): two schemes with comparable watermark strength may still differ in detection efficiency.

Regarding specific methods, mixture of multiple pseudorandom sources will hurt detection efficiency, as more pseudorandom sources means more ambiguity at detection time. (Hu & Huang, 2024) uses one pseudorandom seed. (Dathathri et al., 2024) uses two seeds (draft + target), with multi-round tournaments there may be effectively more. This paper uses three seeds, further compounding interference among signals at detection time. Concretely, for any observed token the detector does not know whether it came from (i) an accepted draft token, (ii) a rejected token replaced from the residual, or (iii) the “extra” token when all draft tokens were accepted. This uncertainty necessarily makes practical detection weaker than the paper’s “WS” metric.

In Experiment section of this paper, the claim is also “improved detectability,” not matching with non-accelerated watermarking.

Therefore, the “trade‑off” still exists. The paper pushes the practical frontier but does not remove it, the headline “BREAK THE TRADE-OFF” is overstated. A more faithful summary is that the method steadily approaches a better trade-off.

### Minors:

The paper, for simplicity, treats positions as independent. A stricter analysis should avoid this assumption, and some results can hold without independence.

Seed reuse is not discussed. Re-applying the same pseudorandom seed should be skipped (e.g., as in prior “context code history”) to maintain unbiasedness.

**Questions:**

I would encourage authors to also report ANLPPT, as with AATPS for efficiency, to quantify the detectability gap relative to a non-speculative watermarked baseline.
How large is the remaining gap in detectable watermark signal strength?

---

> ### Author Response · Authors · 2025-11-21
> **Response to Reviewer**
>
> Thanks for your professional review and positive feedback! The following are our responses:
>
> ### **1. Whether the trade-off truly disappears.**
>
> Thanks for your careful consideration. For a more precise and rigorous presentation, we will change “Break” to “Improve” in the camera-ready version if the paper is accepted. We initially used “Break” in the title to emphasize our theoretical contribution: Algorithm 1 can indeed achieve both maximum watermark strength and maximum sampling efficiency in theory.
>
> ---
>
> ### **2. Skip for repeated context code**
>
> Thanks for reminding us of this point. In our implementation, we skip watermarking for repeated context code, following the approach in [1, 2]. We realize this was not clearly stated in the current presentation, and we have added this implementation detail to Algorithm 1 in the revision.
>
> ---
>
> ### **3. ANLPPT**
>
> Classical p-value–based quantification of detectability does **not directly apply** in our setting because we have two pseudorandom keys. To better illustrate the detectability gap relative to a non-speculative watermarked baseline, we add an **oracle detection curve** to Figures 2 and 3. This curve represents the idealized case in which the detector always selects the correct pseudorandom variable—equivalent to the detection performance of a non-speculative watermark.
>
> ---
>
> ### **References**
> [1] Hu Z, Chen L, Wu X, et al. Unbiased Watermark for Large Language Models[C]//The Twelfth International Conference on Learning Representations.
>
> [2] Hu Z, Huang H. Inevitable trade-off between watermark strength and speculative sampling efficiency for language models[J]. Advances in Neural Information Processing Systems, 2024, 37: 55370-55402.

---

### Official Review · Reviewer_LCLc · 2025-11-02

**Soundness:** 3
**Presentation:** 1
**Contribution:** 3
**Rating:** 6
**Confidence:** 3

**Summary:**

## Summary
This paper tackles a practical tension in LLM serving: **unbiased text watermarking** (for provenance) versus **speculative sampling** (for speed). Prior work framed this as a binary impossibility: you can either preserve the watermarked distribution or preserve the acceptance rate, not both. The authors (i) introduce a **quantitative notion of watermark strength** — the expected KL divergence $E_\zeta[D_{KL}(P_\zeta || P)]$ — and prove it is the **exact detection rate** (via Chernoff–Stein); (ii) cast watermark–efficiency trade-offs as a **Pareto optimization** and empirically map the curve for several watermark classes (Gumbel-max, SynthID, linear mixtures, prior classes); and (iii) propose a simple fix — **pseudorandomizing the accept/reject coin** in speculative sampling — so the whole decode becomes a deterministic function of seeds, achieving **maximum sampling efficiency** $1 - TV(Q,P)$ and **maximum watermark strength** $H(P)$ simultaneously. They also give practical detectors for speculative pipelines (Ars-$\tau$ / Ars-Prior and Bayes-MLP / Bayes-Prior).

Production stacks already depend on speculative sampling; having a *measurable* frontier and a concrete recipe to sit on that frontier is valuable.

## Main claims / contributions
1. **Strength as information:** Define watermark strength as $E_\zeta[D_{KL}(P_\zeta || P)]$ and prove it controls the optimal detector’s p-value decay (through the per-token log-likelihood ratio).
2. **Tight frontier & optima:** For unbiased schemes, strength is upper-bounded by $H(P)$; speculative acceptance is upper-bounded by $1 - TV(Q,P)$. Gumbel-max and SynthID (as $m \to \infty$) are strength-optimal; the proposed **seeded acceptance** achieves *both* bounds simultaneously.
3. **Operational trade-offs & detectors:** A constrained program (Eq. 10) traces the **Pareto curve** $L(r)$. The paper designs detectors that account for draft/target paths; when the accept coin is known (Ars-$\tau$, Bayes-MLP), detection is stronger than prior averaging approaches (Ars-Prior, Bayes-Prior).

**Strengths:**

## Strengths
- **Conceptual clarity:** Moving from a binary notion to a continuous, information-theoretic strength is the right abstraction. It cleanly explains earlier "impossibility" statements and yields a usable frontier.
- **Tight theory that lines up with practice:** The Chernoff–Stein interpretation makes sample-complexity predictions directly actionable; the entropy and TV bounds give interpretable ceilings.
- **Simple, impactful algorithmic tweak:** Seeding the accept coin is minimal and compatible with production speculative pipelines.
- **Detector design under speculative sampling:** The draft/target path issue is real; the two Ars and two Bayes variants are a neat, pragmatic treatment.
- **Figures are helpful:** The trade-off plots communicate the Pareto picture at a glance; markers (WS/SE maintained, optimum) guide interpretation.

**Weaknesses:**

## Weaknesses / limitations
- **Oracle knowledge vs deployment reality:** Many results assume access to the base distribution $P_t$ (or accurate logits) and, for the strongest detectors, the **accept coin**. External forensics often lacks both; estimation error can materially affect strength and false positives.
- **Model/scale realism:** Simulated $(Q,P)$ pairs and small-vocab settings may not capture calibration quirks, long-tail tokens, or beam-blocking in large models.
- **Robustness / attack model:** The paper doesn’t stress-test adversaries who (i) alter spacing/punctuation, (ii) paraphrase with another model, or (iii) run ensemble-style removal — how quickly does the effective KL collapse?
- **Broader behavioral impacts not discussed:** Recent studies suggest watermarking can alter model behavior beyond detectability or quality trade-offs. *Downstream Trade-offs of a Family of Text Watermarks* (Ajith et al., EMNLP 2024) reports 10–20% drops in downstream task accuracy even for “unbiased” schemes like KGW, while *WaterJudge* (Molenda et al., NAACL 2024) quantifies a detectability–quality trade-off across watermark strengths. *Watermarking Degrades Alignment in Language Models* (Verma et al., ICLR 2025) further shows that watermarking can shift alignment properties such as truthfulness, safety, and helpfulness, mitigated partly by an external reward model. The paper would be stronger with a targeted discussion of how speculative sampling with watermarking affects alignment-relevant metrics (e.g., reward scores) and a short discussion citing these works to contextualize potential side effects.
- **Downstream behavior:** Stronger coupling can interact with alignment/quality. Recent work reports reward/quality/alignment degradations under watermarking; the paper doesn’t quantify the proposed method’s effect on reward score or human preference.
- **Reporting details:** Some derivations (e.g., Eq. 10) are explained tersely; detector training details and calibration of $\tau$ / $p$ could be easier to reproduce.

## Actionable suggestions
1. **Adversarial/transform robustness:** Evaluate detectability after common edits (paraphrase via another model, synonym swaps, punctuation/whitespace jitter, mild re-ordering). Report the induced drop in effective strength and how close it is to theoretical predictions.
2. **Latency & throughput accounting:** Since the goal is "fast **and** provable," include an end-to-end latency/throughput table under seeded acceptance (overhead of PRNG, acceptance calibration, detector extraction) to confirm the practical speedup survives.
3. **Downstream effects audit:** Report reward-model scores or human preference deltas for seeded-accept decoding vs baseline speculative and vs classic watermarks. Even a small ablation will reassure readers that “max strength” doesn’t quietly harm alignment/quality.
4. **Clarity & reproducibility:** Expand the derivation around Eq. 10 in the main text (one boxed paragraph). Right now it jumps out of the blue without providing much background.

The narrative is coherent and the prior binary impossibility is treated respectfully. I appreciated the clean placement of this work relative to Hu & Huang and SynthID, as well as the detector connection to Bayesian scoring networks. I would like to see clearer statements about assumptions (knowledge of $P_t$, seed/coin access) up front.


## Writing and Presentation
Generally clear but somewhat difficult to follow; figures carry their weight. The "what to believe after reading" is crisp: *use KL as strength, plot the Pareto, seed the accept coin to sit on the frontier.* A bit more exposition around the constrained program and detector calibration would help readers who don’t already know speculative internals.

## Nits / small issues
- Define “accept coin” once in the intro or prelims to avoid hunting later.
- Around the entropy upper bound, one sentence explaining $WS = H(P) - E[H(P_\zeta)]$ would help a beginner.
- Minor typographical cleanups: consistent capitalization of "SynthID," spacing around $TV(\cdot,\cdot)$, and equation referencing.

**Questions:**

See above

---

> ### Author Response · Authors · 2025-11-21
> **Response to Reviewer**
>
> Thanks for your professional review and positive feedback! The following are our responses:
>
> ### **1. Oracle knowledge vs deployment reality**
>
> Thanks for raising this concern. We would like to clarify the reviewer’s misunderstanding: we assume access to the true token distribution **only in the theoretical analysis of watermark strength**, not in our experimental evaluation. In the experiments, we do **not** rely on such knowledge.
>
> ---
>
> ### **2. Model/scale realism**
>
> Thanks for raising this concern. The simulation experiment in Section 3.2 is designed **purely to illustrate and visualize the trade-off curve** under our definition of watermark strength. To keep this illustration clear and interpretable, we intentionally use *reasonablely simplified* $(Q, P)$ pairs (See Appendix C.1 for more discussion). Incorporating model-specific calibration effects, long-tail distributions, or beam-blocking behaviors would unnecessarily complicate the simulation, risk confusing readers, and is not needed for the purpose of illustrating the theoretical trade-off.
>
> Importantly, Section 5 provides **comprehensive experiments in realistic LLM settings**, using full-scale draft–target model pairs and real-world datasets. The results demonstrate that our method **aligns well with theoretical analysis** under such realistic conditions.
>
> ---
>
> ### **3. Robustness / attack model**
>
> Thanks for raising this point. Our work *focuses on characterizing and improving the trade-off between watermark strength and sampling efficiency* rather than exhaustive empirical benchmarking. Thus current experiments are sufficient to support our main contributions. Robustness evaluation is beyond the scope of our main contributions. That said, we have already listed the robustness evaluation as a limitation, and we agree it represents an important direction for future work.
>
> ---
>
> ### **4 & 5. Broader behavioral impacts not discussed**
>
> Thanks for highlighting recent studies on broader behavioral impacts of watermarking. Our work *focuses on characterizing and improving the trade-off between watermark strength and sampling efficiency* and these listed works are *not* directly relevant to our work. These works examine how specific watermarking **schemes** (e.g., KGW or reweight-based methods) may affect *downstream accuracy*, *alignment*, or *reward-model scores*. These broader behavioral metrics (alignment, safety, task accuracy) fall outside the scope of our main contributions.
>
> ---
>
> ### **6. Latency & throughput accounting**
>
> Thanks for this suggestion. We now report **Per Token Time (PTT)** in the updated *Table 1*, following the reviewer’s suggestion. The results show that, compared with basic watermarking, our method (watermarking with speculative sampling and pseudorandom acceptance) does **improve sampling speed**.
>
> We also want to clarify the distinction between **theoretical speedup** and **empirical runtime**. In Table 1, Average Accepted Tokens Per Step (AATPS) measures the number of accepted tokens in each generation loop and thus reflects the theoretical speedup, focusing primarily on the draft token acceptance rate, to which our method **directly contributes**. In contrast, PTT empirically measures the actual runtime of generation and can be affected by *various factors*, including (but not limited to) watermark sampling, token verification, and model switching (when using a single GPU). Therefore, the observed speedup based on PTT may not perfectly align with that implied by AATPS, and further runtime analysis lies outside the scope of our main contributions.
>
> ---
>
> ### **7. More details**
>
> Thanks for the great suggestions for improving the presentation. The following is our plan for revision:
>
> 1. We added more details for the derivation around Eq. 10 in the main text.
> 2. We added additional details about the detector training and calibration procedures for both Gumbel-max and SynthID to improve reproducibility.
> 3. We added a note below the definition of watermark strength to indicate that it can be interpreted as the conditional KL divergence (which, under unbiasedness, equals the mutual information $I(W;\zeta)$). In addition, in Theorem 3.2, we clarify that, under unbiasedness, the watermark strength can also be written as $H(P) - E_{\zeta}[H(P_{\zeta})]$. This should help readers with an information-theoretic background connect our definition to standard concepts.
> 4. We will carefully check all mathematical expressions and their references for clarity and correctness.

---

### Official Review · Reviewer_m26b · 2025-11-03

**Soundness:** 4
**Presentation:** 4
**Contribution:** 4
**Rating:** 8
**Confidence:** 3

**Summary:**

This paper considers watermarking when speculative sampling is used. Their results seem modular, allowing for improved watermarking with speculative sampling by improving standard watermarking schemes and applying them to both the draft and target models. Making this optimal seems to require an interesting trick that (pseudo)randomizes acceptance. They appear to understand all the key points, like that this will not introduce new distortions if the watermarks for the two models are independent.

**Strengths:**

The paper has several strong results. The basic observation, that one can overcome the limitation of Hu et al. by allowing for pseudorandom draft-token acceptance, is very nice and important. Watermarking under speculative sampling is an extremely important technical issue for deploying watermarks in real-world LLMs.

I haven't read the paper carefully, but it seems to make real progress on an important practical question.

**Weaknesses:**

I don't know of any.

**Questions:**

Is Google's watermark really a deterministic function of pseudorandom numbers? I thought that tournament sampling doesn't satisfy this property at all: Doesn't it work by first sampling a few tokens, and then choosing deterministically between them? In which case, there's still a bunch of entropy left on the table in the initial sampling process.

---

> ### Author Response · Authors · 2025-11-21
> **Response to Reviewer**
>
> Thank you for your positive feedback—we truly appreciate your encouraging comments. The following are our responses:
>
> ### **Explanation for SynthID**
>
> Thanks for raising this question. When the tournament sampling round $m$ is finite, the sampling process remains stochastic (non-deterministic function of pseudorandom variable). However, as $m \to \infty$, the distribution **converges almost surely** to a degenerate one, meaning the output becomes deterministic. This is the sense in which SynthID achieves maximal watermark strength in our theoretical analysis. The proof for the limiting distribution is provided in Appendix B.3.

---

### Official Review · Reviewer_4VEY · 2025-11-08

**Soundness:** 3
**Presentation:** 3
**Contribution:** 3
**Rating:** 6
**Confidence:** 4

**Summary:**

This paper studies the trade-off between watermarking and speculative decoding. A relatively recent work had made a 'no-free lunch' observation, which shows that improving the watermark is at the cost of weakening the speculative decoding (acceptance rate), and vice versa. This paper challenges this observation by integrating the watermarking's pseduorandom mechanism into the speculative decoding mechanism. The paper studies the speculative-decoding and watermarking tradeoff and the resulting curves, proposes a method to 'break' these curves and presents some experiments, demonstrating the proposed method on the Gumbel and SynthID watermarks.

**Strengths:**

Overall, this is a solid and creative work. Here are some of its strengths:

1. The paper is clearly written. Even though I have significant prior knowledge about both watermarking and speculative decoding I believe this paper will not be too hard to follow for a newcomer to these fields.

2. The idea is novel and seemingly powerful. Furthermore, the authors address the combination of two important subjects in the AI community - trustworthiness and efficiency.

3. The analysis is very interesting and creative! Reading this paper was really delightful.

**Weaknesses:**

### Major:

- **Clarity on SynthID:** The paper uses SynthId as a case-study in this work, as it is an 'unbiased watermark'. However, SynthID is a relatively large class of watermarks that follow the tournament sampling mechanism. The general SynthID watermark is not even unbiased (when N>2). Unfortunately, the authors do not provide sufficient information to understand which specific case of SynthID is proposed in this work. I also believe that such information should be added for the sake of clarity.

- **Missing related work:** The content of this paper misses several related works on 'unbiased' watermarks for better context, which can also be considered for the analysis and experiments. The first [1] considers a similar hypothesis test to the one that is considered int his work, however through the Bayesian setting. This work is highly relevant due to its analysis in the 'token-level', as considered in this work. Furthermore, the results of [1] are stated in terms of the total variation distance, which may lead to easier integration with the speculative-decoding analysis that is done in this work. Beyond that work, there are highly related work that is missing for context - [2] provides a general formulation in terms of score and watermark transform optimization, proposing methods that outperform the considered watermarks in this work. [3] studies optimality

- **Experiments section should be extended.** I would expect this section to consider more experiments, through more datasets (testing across specific tasks/domains) and testing more aspects of watermarking (robustness?). The current collection is quite narrow. Furthermore, I would expect a quantification of the watermark strength through p-values and quantification of the bias (through either textual quality or cross entropy/relative perplexity).

### Minor:
- The title of section 3 "COMPLETE THE TRADE-OFF DIAGRAM" is not clear.

[1] Tsur, D., Long, C. X., Verdun, C. M., Hsu, H., Permuter, H., & Calmon, F. P. (2025). Optimized Couplings for Watermarking Large Language Models. arXiv preprint arXiv:2505.08878.‏

[2] Tsur, D., Long, C. X., Verdun, C. M., Vithana, S., Hsu, H., Chen, C. F., ... & Calmon, F. HeavyWater and SimplexWater: Distortion-free LLM Watermarks for Low-Entropy Distributions. In The Thirty-ninth Annual Conference on Neural Information Processing Systems.‏‏

[3] He, Haiyun, et al. "Universally optimal watermarking schemes for llms: from theory to practice." (2024).‏

**Questions:**

Bellow are both questions and comments:

- **Connection to (Hu & Huang 24) and the hypothesis test:** The difference from the results in (Hu & Huang 24). which is explained in Remark 3.1 is unclear to me. First, the comparison itself is a bit odd. The referred paper discusses asymptotic results of the hypothesis test, while this work (while not stating clearly) is a token-level mechanism. The notion of a sequence is not addressed at any stage of this work. Second, Both works quantify the watermarks strength (detection) through the KL between distributions. This is an immediate results from the error exponent of the hypothesis test.

- Following the previous comment, as the authors consider the 'token-level' hypothesis test, the use of the KL divergence to quantify the distributional divergence seems a little odd. In the hypothesis testing context, and as noted in this paper, KL divergence emerges in an asymptotic analysis (Through the error exponent of the type-2 error in the equivalent independence the between the 'i.i.d.' tokens and the psuedorandom process). The setting that manifests the KL divergence doesn't seem to fit strongly in the proposed operational setting of this work. I would appreciate the authors' thoughts on that. I believe that the mentioned related works [1,2] might be highly impactful to address this issue.

- The expected KL divergence in eqn (7) is a popular quantity in information theory termed 'conditional KL divergence' - I would use that standard notation.

- I believe that the watermark transformation can be regarded as a transition kernel similarly to the speculative decoding mechanism. Is there a reason to not address is as one in this work?

- Do the results of this work extend to the unbiased case? I would believe that the tilting mechanism in [2, Section 2] might come in handy as a formulation of 'removing bias'.

- Is the proposed method applicable to a general new watermark or does a new solution should be tailored to each new watermark?

---

> ### Author Response · Authors · 2025-11-21
> **Response to Reviewer (part A)**
>
> Thanks for your professional review and positive feedback! The following are our responses:
>
> ### **1. Clarity on SynthID**
>
> Thanks for pointing this out. In our work, we use the **unbiased two-candidate version** of SynthID—that is, we set **$N = 2$** and use Bernoulli(0.5) as the $g-$value following the default setting in SynthID paper [1]. We agree that this should be stated more explicitly, and we have *added* a brief clarification of this in the *Preliminaries* section when introducing SynthID.
>
> Regarding the number of tournament rounds:
>
> - For **theoretical analysis** (e.g., in establishing maximal watermark strength), we set **$m = \infty$**, unless otherwise specified (clarified around *line 307*).
> - For **experiments**, we follow the practical setting used in [1] and set **$m = 30$** (clarified around *line 454*).
>
> ---
>
> ### **2. Missing related work**
>
> We thank the reviewer for pointing out these interesting related works. **We will cite these works and add a brief discussion in the revised related work section**. The three papers mentioned provide valuable context on unbiased watermarking: (1) *Optimized Couplings for Watermarking LLMs*  analyzes one-shot watermarking under a hypothesis-testing framework and characterizes detection–perception trade-offs; (2) *HeavyWater and SimplexWater*  introduce optimization-based, distortion-free watermarks that can outperform classical schemes; and (3) *Universally Optimal Watermarking Schemes*  jointly optimize watermarking and detection under worst-case Type-I error control.
>
> The above papers focus on *designing* stronger unbiased watermarks, whereas our framework applies to unbiased watermarks and focuses on their interaction with speculative decoding. Still, we agree that these works provide useful context.
>
> ---
>
> ### **3. Experiments section should be extended**
>
> Thanks for the great suggestions. The following are our clarifications for more experiments.
>
> 1. **Evaluate on more datasets.** We believe the current experiments are sufficient to support our main contributions. Our method **functions the same way** across datasets, as its mechanism does not rely on any assumptions about next-token prediction. For this reason, using the widely used **ELI5 dataset** is sufficient to validate our method. *That said, we have also conducted experiments on an additional dataset (C4 dataset) and have added the results in Appendix F.2.* These additional results are *consistent* with our existing findings and *further validate* our method.
>
> 2. **Evaluate on more aspects (robustness).** Our work *focuses on characterizing and improving the trade-off between watermark strength and sampling efficiency* rather than exhaustive empirical benchmarking. Thus, current experiments are sufficient to support our main contributions. Robustness evaluation is beyond the scope of our main contributions. That said, we have already listed the robustness evaluation as a limitation, and we agree it represents an interesting direction for future work.
>
> 3. **Quantification of watermark strength.**
>
>    We would like to clarify a key misunderstanding. In our work, watermark strength (Definition 3.1) measures the degree to which the watermarked token depends on, or is coupled with, the pseudorandom numbers. In contrast, the p-value is a quantity used to evaluate detectability, namely the Type II error at a fixed Type I error level. These concepts are fundamentally different: watermark strength is defined under the assumption that the NTP distributions are known, whereas practical detectability concerns the scenario where these distributions are unknown. This distinction is precisely what Remark 3.1 aims to convey.
>
>    We speculate that the reviewer is suggesting the use of p-values as an alternative perspective for studying detectability. However, in practice, computing p-values is difficult because the null distribution of the score is generally unknown. For this reason, we instead **present ROC curves** to illustrate detectability (see Figure 4 in the Appendix). In fact, the ROC curve fully characterizes detectability and can be used to derive p-values if the null distribution becomes available.
>
>
> 5. **Evaluating bias/text quality.** We now report **log-perplexity** in the updated *Table 1*, following the reviewer’s suggestion. The results show that our method—including pseudorandom acceptance—does **not degrade text quality** compared to standard speculative sampling. More broadly, text quality can be influenced by multiple components (speculative sampling strategy, watermarking scheme, draft–target model choice, etc.), and a deeper study is beyond the scope of this work. We have also listed text-quality impacts as part of the future work (see *line 500*).
>
> 6. **Refine the section 3 title.** We have changed the title to *“Complete the Trade-off Curve”* to improve readability. Thanks for this suggestion.

---

> ### Author Response · Authors · 2025-11-21
> **Response to Reviewer (part B)**
>
> ### Responses to Questions:
>
> 1. **Remark 3.1 and KL divergence.**
>
>    We would like to clarify a misunderstanding: Remark 3.1 does not discuss Hu & Huang’s work. Instead, it explains the conceptual distinction between the notion of watermark strength introduced in our work and the detection efficiency studied in Li et al. [1]. We agree with the reviewer that Li et al. [2025] adopt an asymptotic viewpoint where the token length grows to infinity. However, the aspect of their work that we wish to highlight is different: their efficiency notion characterizes the worst-case decay rate of the Type II error, under a fixed Type I error, when token distributions range over a prior class. This notion directly reflects practical detectability and explicitly accounts for the fact that the NTP distributions are unknown.
>
>    Our notion is also asymptotic, but with a key difference: we do not take the worst-case perspective, and we instead examine the decay of the Type I error. Consequently, our definition implicitly assumes knowledge of the NTP distributions. This is intentional—watermark strength is designed as a theoretical quantity that measures how strongly tokens depend on pseudorandom numbers, and is therefore idealized by construction.
>
>    Remark 3.1 is intended precisely to clarify this conceptual distinction.
>
> 2. **Conditional KL divergence.** Thanks for pointing this out. We have now *added* a note to Definition 3.1 indicating that the expected KL divergence can also be interpreted as the conditional KL divergence (and, under unbiasedness, equals the mutual information $I(w;\zeta)$). This should help readers familiar with information theory better connect our definition to standard concepts.
> 3. **Watermark as a transition kernel.** In our paper, our goal was **not** to emphasize the internal mechanics of each watermarking scheme, so we did *not explicitly* express the watermark as a transition kernel in the same form as the speculative-sampling kernel in Eq. (5). Also, we follow the traditional way used in previous papers [1-5] to present the watermark. That said, the reviewer’s interpretation is correct: watermark transformation can indeed be viewed as a transition kernel. Thanks for raising this point and discussing with us.
> 4. **Extend to unbiased case**  *The two watermarking schemes studied in our work—Gumbel-max and SynthID—are both unbiased*. The proposed method can be directly applied to **any unbiased watermark** that already achieves **maximum watermark strength** (as characterized in Definition 3.1 and Theorem 3.2), which requires the watermarked distribution to be **degenerate**—i.e., all probability mass concentrated on a single token. Based on our understanding of [2], both HeavyWater and SimplexWater are unbiased but **do not guarantee degenerate outputs**, and therefore do not attain maximal watermark strength. While our method can still be applied to these schemes, the resulting watermark strength would no longer reach the theoretical maximum. Also, to our understanding, the tilting mechanism in [6, Section 2] sacrifices the unbiasedness to gain more detectability, so the watermark is no longer unbiased and can not be applied to our method. If we have misunderstood any aspect of this mechanism, we welcome clarification.
> 5. **New solution to new watermark.** We interpret the reviewer’s question about a “new solution” as referring to **new detection methods**. In our framework, the detector depends on the specific structure of the watermarking scheme. Therefore, for a new watermark, the **detection rule must be adapted accordingly**.
>
> ---
>
> ### **References**
>
> [1] Dathathri, Sumanth, et al. "Scalable watermarking for identifying large language model outputs." *Nature* 634.8035 (2024): 818-823.
>
> [2] Li, Xiang, et al. "Robust detection of watermarks for large language models under human edits." *Journal of the Royal Statistical Society Series B: Statistical Methodology* (2025): qkaf056.
>
> [3] Kuditipudi, Rohith, et al. "Robust distortion-free watermarks for language models." arXiv preprint arXiv:2307.15593 (2023).
>
> [4] Li, Xiang, et al. "A statistical framework of watermarks for large language models: Pivot, detection efficiency and optimal rules." The Annals of Statistics 53.1 (2025): 322-351.
>
> [5] Hu, Zhengmian, et al. "Unbiased watermark for large language models." *arXiv preprint arXiv:2310.10669* (2023).
>
> [6] Tsur, D., Long, C. X., Verdun, C. M., Vithana, S., Hsu, H., Chen, C. F., ... & Calmon, F. HeavyWater and SimplexWater: Distortion-free LLM Watermarks for Low-Entropy Distributions. In The Thirty-ninth Annual Conference on Neural Information Processing Systems.

---

> > ### Comment · Reviewer_4VEY · 2025-11-27
> >
> > I appreciate the author's detailed response to my review and the clarification to the points I addressed.
> >
> > - I indeed had a minor typo in my question about extension of this scheme (one before last) - I meant to ask if the authors expect this method to extend to the **biased** case, in which we potentially sacrifice quality for better detection. I think that practically, accompanying 'softer' solutions that rely on distribution reweighing rather than only focusing on mechanisms that result in 'degenerate distributions' might be beneficial for the overall applicability of this work. I do not expect any changes to be made in the paper within this period but discussing non-degenrate solutions and possible extensions to biased solutions might help to broaden the limits of this work.
> >
> > Finally, following my discussion with the authors and their effort to address my comments, I have decided to raise my score.

---

> > > ### Author Response · Authors · 2025-11-27
> > >
> > > Thank you for clarifying your comment and for raising the score. We agree that extending our framework and establishing similar improvements for non-degenerate and even biased watermarks is an interesting and promising future direction.
> > >
> > > We *have added a discussion* regarding this extension in Section 6 and believe it will help inspire future work in this area.

---

### Author Response · Authors · 2025-11-26
**Appreciation and Welcome Further Discussions**

We want to thank all the reviewers for your dedicated efforts and positive feedback on our manuscript. In the current revision, we have added necessary clarifications and additional experiments, which are highlighted in *purple*.

As the rebuttal deadline is approaching, we are eager to address any remaining questions that you may have. We welcome any additional comments or suggestions for clarification and are prepared to conduct further experiments if needed. We look forward to your guidance to help us improve our paper further.

---

### Author Response · Authors · 2025-12-03
**Rebuttal Summary (part I)**

**To the Area Chair:**

We sincerely thank you for your time and extra effort in overseeing the review process.

Overall, the reviewers provided *positive feedback* on our work. During the rebuttal, we carefully addressed every concern raised by the reviewers and added the necessary additional experiments to clarify key points. Below is a more detailed summary of our responses for your convenience. If you would like further details, please follow the hyperlinks, which lead to the corresponding specific discussion.

- **[Reviewer 4VEY](https://openreview.net/forum?id=HA8vzzT6Ax&noteId=5D2niQL2pU).** As noted by the reviewer, *we have addressed all concerns, and the reviewer has raised their score to 8 before the score revert ([link](https://openreview.net/forum?id=HA8vzzT6Ax&noteId=BRGYdckHo6))*. The reviewer’s major concerns were: (1) clarity regarding SynthID, (2) missing related work, and (3) the need for additional experiments. We addressed them as follows:
  - We added a clear clarification of the specific SynthID variant used in our work.
  - We added the suggested related works along with a brief discussion.
  - We conducted experiments on an additional dataset and also reported **log-perplexity** for text quality evaluation; the new results are *consistent* with our existing findings and *further validate* our method.
  - Additionally, we addressed all other questions raised by the reviewer and refined the revision accordingly.
- **[Reviewer m26b](https://openreview.net/forum?id=HA8vzzT6Ax&noteId=DN4JcSSNY1).** The reviewer provided *very positive feedback* and raised a single question regarding the determinism of the SynthID watermark. In the rebuttal, we offered a clear clarification and pointed the reviewer to the corresponding explanation in the appendix.
- **[Reviewer LCLc](https://openreview.net/forum?id=HA8vzzT6Ax&noteId=VD6SypuVQi).** The reviewer's concern can be summarized into four points: (1) deployment realism and assumptions, (2) robustness against attacks and broader impact on downstream tasks, (3) empirical runtime reporting, and (4) presentation details. We addressed these as follows:
  - We clarify the reviewer's *misunderstanding*: we assume access to the true token distribution **only in the theoretical analysis of watermark strength**, not in our experimental evaluation. In the experiments, we do **not** rely on such knowledge. Also, we clarify that the simulation experiment in Section 3.2 is designed **purely to illustrate and visualize the trade-off curve** under our definition of watermark strength; introducing additional complexities would not meaningfully contribute to this purpose.
  - We emphasize that our work *focuses on characterizing and improving the trade-off between watermark strength and sampling efficiency* rather than exhaustive empirical benchmarking. Thus, current experiments are *sufficient* to support our main contributions. Robustness evaluation under attack and broader behavioral impact are *beyond* the scope of our main contributions.
  - We *reported* **Per Token Time (PTT)** in the updated *Table 1, 2*, following the reviewer’s suggestion. The results show that, compared with basic watermarking, our method (watermarking with speculative sampling and pseudorandom acceptance) does **improve sampling speed**. Additionally, we *added* a discussion regarding the difference between **theoretical speedup** and **empirical runtime** in the appendix.
  - We refined presentation throughout the paper and *added* the corresponding clarifications in the revision.
- **[Reviewer d8rs](https://openreview.net/forum?id=HA8vzzT6Ax&noteId=bnG7cdJsOb).** The reviewer provided *very positive feedback* and suggested the following improvements: (1) reconsider the word “Break” in the title; (2) skip the reused pseudorandom seed; and (3) report ANLPPT to quantify the detectability gap. We adopted these suggestions as follows:
  - For a more precise and rigorous presentation, we will change “Break” to “Improve” in the camera-ready version if the paper is accepted.
  - We *already skipped* the reused pseudorandom seed on our work, but didn't mention this in our original version. We *added* this implementation detail in the revision.
  - We clarify that classical p-value–based quantification of detectability does **not directly apply** in our setting because of the ambiguity caused by two pseudorandom keys. To better illustrate the detectability gap relative to a non-speculative watermarked baseline, we *added* an **oracle detection curve** to Figures 2, 3, 5, 6 in the revision.

---

> ### Author Response · Authors · 2025-12-03
> **Rebuttal Summary (part II)**
>
> - **[Reviewer sUKk](https://openreview.net/forum?id=HA8vzzT6Ax&noteId=fYIWbJdugf).** The reviewer raised two concerns: (1) determinism of the SynthID watermark and (2) reliance on strong assumptions. We addressed them as follows:
>   - We provided a clear clarification of when SynthID behaves stochastically versus deterministically and directed the reviewer to the corresponding appendix section.
>   - We clarify that these assumptions we made are *not* idealized or unrealistic—they are **standard in the unbiased watermarking literature** and are **achievable in practical LLM decoding**. In the rebuttal, we provided *detailed explanations together with examples* regarding the reasonableness of assuming unbiased decoders, fully deterministic generation, and independent pseudorandom seeds. Additionally, our experiments show consistent results with theoretical analysis, which further suggests that our method delivers the intended trade-off improvements under realistic and general conditions.
> - **[Reviewer Zd3w](https://openreview.net/forum?id=HA8vzzT6Ax&noteId=PCJTNKWyr7).** The reviewer raised three concerns: (1) extend to other unbiased watermarks, (2) analyze the influence on text quality, and (3) empirical runtime reporting. We addressed them as follows:
>   - We emphasize that our focus is on *characterizing and improving the trade-off between watermark strength and sampling efficiency* rather than exhaustive empirical benchmarking for various watermarks. We believe the current experiments are sufficient to support our main contributions. Additionally, we clarify what watermark can be used for our framework and explain the reason.
>   - We reported **log-perplexity** in the updated *Table 1, 2* to evaluate the text quality, following the reviewer’s suggestion. The results show that our method—including pseudorandom acceptance—does **not degrade text quality** compared to standard speculative sampling.
>   - We reported **Per Token Time (PTT)** in the updated *Table 1, 2*, following the reviewer’s suggestion. The results show that, compared with basic watermarking, our method (watermarking with speculative sampling and pseudorandom acceptance) does **improve sampling speed**.

---

### Meta-Review · Area_Chair_nPDn · 2026-01-05

**Summary:**

Reviewers broadly agree that the paper makes a strong and timely contribution by reframing the watermarking–speculative sampling trade-off through an information-theoretic notion of watermark strength and providing a principled mechanism to improve the trade-off substantially. The theoretical analysis is viewed as clean and insightful, and the proposed pseudorandom acceptance mechanism is elegant and practically relevant for real LLM deployment.

While some concerns were raised about assumptions, scope of experiments, and terminology, the overall consensus is that the paper clears the ICLR bar.

**Reviewer Concerns:**

Most reviewer concerns were adequately addressed in the rebuttal, including clarifications on SynthID variants, assumptions behind theoretical optimality, additional related work, runtime reporting, and text-quality evaluation. Remaining concerns primarily relate to broader robustness, downstream behavioral impacts, and extensions to biased or non-degenerate watermarking schemes, which reviewers generally agreed are reasonable limitations and future directions rather than blockers. The authors’ responses and revisions sufficiently resolve the core technical questions raised during review.

**Reviewer Scores:**

Reviewer 4VEY raised their score after discussion and would likely remain clearly positive. Reviewers m26b and d8rs were already strong accepts and would maintain their scores. Reviewers LCLc, sUKk, and Zd3w would likely remain marginal-to-clear accepts, given that their main concerns were addressed or appropriately scoped.

---

### Decision · Program_Chairs · 2026-01-26

Accept (Poster)